# Binding of *LncDACH1* to dystrophin impairs the membrane trafficking of Nav1.5 protein and increases ventricular arrhythmia susceptibility

Genlong Xue[1,2†], Jiming Yang[1†], Yang Zhang[1†], Ying Yang[1], Ruixin Zhang[1], Desheng Li[1], Tao Tian[1], Jialiang Li[1], Xiaofang Zhang[1], Changzhu Li[1], Xingda Li[1], Jiqin Yang[1], Kewei Shen[1], Yang Guo[1], Xuening Liu[1], Guohui Yang[1], Lina Xuan[1], Hongli Shan[3], Yanjie Lu[1], Yang Baofeng[1,4*], Zhenwei Pan[1,5*]

[1]Department of Pharmacology (The Key Laboratory of Cardiovascular Research, Ministry of Education) at College of Pharmacy, Harbin Medical University, Harbin, China; [2]The Institute of Heart and Vascular Diseases, Department of Cardiology, and Central Laboratory, the First Affiliated Hospital of Dalian Medical University, Dalian, China; [3]Shanghai Frontiers Science Research Center for Druggability of Cardiovascular noncoding RNA, Institute for Frontier Medical Technology, Shanghai University of Engineering Science, Shanghai, China; [4]Research Unit of Noninfectious Chronic Diseases in Frigid Zone, Chinese Academy of Medical Sciences, Beijing, China; [5]NHC Key Laboratory of Cell Transplantation. The First Affiliated Hospital of Harbin Medical University, Harbin, China

**\*For correspondence:**
yangbf@ems.hrbmu.edu.cn (YB);
panzw@ems.hrbmu.edu.cn (ZP)

[†]These authors contributed equally to this work

**Competing interest:** The authors declare that no competing interests exist.

## eLife assessment

This study presents an **important** contribution to cardiac arrhythmia research by demonstrating long noncoding RNA Dachshund homolog 1 (lncDACH1) tunes sodium channel functional expression and affects cardiac action potential conduction and rhythms. The evidence supporting the major claims are **convincing**. The work will be of broad interest to cell biologists and cardiac electrophysiologists.

**Abstract** Dystrophin is a critical interacting protein of Nav1.5 that determines its membrane anchoring in cardiomyocytes. Long noncoding RNAs (lncRNAs) are involved in the regulation of cardiac ion channels, while their influence on sodium channels remains unexplored. Our preliminary data showed that lncRNA-*Dachshund* homolog 1 (*lncDach1*) can bind to dystrophin, which drove us to investigate if *lncDach1* can regulate sodium channels by interfering with dystrophin. Western blot and immunofluorescent staining showed that cardiomyocyte-specific transgenic overexpression of *lncDach1* (*lncDach1*-TG) reduced the membrane distribution of dystrophin and Nav1.5 in cardiomyocytes. Meanwhile, peak $I_{Na}$ was reduced in the hearts of *lncDach1*-TG mice than wild-type (WT) controls. The opposite data of western blot, immunofluorescent staining and patch clamp were collected from *lncDach1* cardiomyocyte conditional knockout (*lncDach1*-cKO) mice. Moreover, increased ventricular arrhythmia susceptibility was observed in *lncDach1*-TG mice in vivo and ex vivo. The conservative fragment of *lncDach1* inhibited membrane distribution of dystrophin and Nav1.5, and promoted the inducibility of ventricular arrhythmia. Strikingly, activation of *Dystrophin* transcription by dCas9-SAM system in *lncDach1*-TG mice rescued the impaired membrane distribution of dystrophin and Nav1.5, and prevented the occurrence of ventricular arrhythmia. Furthermore, *lncDach1* was increased in transaortic constriction (TAC) induced failing hearts, which promoted

the inducibility of ventricular arrhythmia. And the expression of *lncDach1* is regulated by hydroxya-cyl-CoA dehydrogenase subunit beta (hadhb), which binds to *lncDach1* and decreases its stability. The human homologue of *lncDACH1* inhibited the membrane distribution of Nav1.5 in human iPS-differentiated cardiomyocytes. The findings provide novel insights into the mechanism of Nav1.5 membrane targeting and the development of ventricular arrhythmias.

## Introduction

The voltage-gated sodium channel mediates the 0 phase depolarizing inward sodium currents of cardiomyocytes (*Rook et al., 2012*). The expression and function of the sodium channel is regulated at multiple levels encompassing gene mutation, post-transcriptional modification, post-translational modification, and protein trafficking etc (*Rook et al., 2012*; *Marionneau and Abriel, 2015*). The disruption of either process is arrhythmogenic and occasionally causes sudden death (*Marionneau and Abriel, 2015*; *Ruan et al., 2009*).

The membrane targeting and localization of the pore-forming subunit of sodium channel Nav1.5 was regulated by several interacting proteins such as ankyrin-G, MOG1, syntrophin, and dystrophin etc (*Marionneau and Abriel, 2015*; *Abriel et al., 2015*). Dystrophin is an intracellular protein that is encoded by the *duchenne muscular dystrophy* (*DMD*) gene (*Lapidos et al., 2004*). It distributes mainly on the cellular membrane of skeletal muscle cells and cardiomyocytes, and acts as a scaffold for Nav1.5 (*Gavillet et al., 2006*). In cardiomyocytes, *Dmd* controls the expression and membrane anchoring of Nav1.5. *Gavillet et al., 2006* showed that knockout of dystrophin in cardiomyocytes reduced peak sodium current, Nav1.5 protein expression, and conduction velocity in mice. Subsequently, they confirmed that the knockout of dystrophin inhibits membrane distribution of Nav1.5 due to the disruption of dystrophin-syntrophin complex (*Petitprez et al., 2011*).

LncRNAs are a new class of RNAs that are more than 200 nts long and possess little protein-coding property. (*Quinn and Chang, 2016*) LncRNAs have been shown to regulate multiple biological processes and participate in the pathogenesis of various diseases including cardiac diseases (*Bär et al., 2016*). LncRNAs were shown to regulate cardiac electrophysiological properties by altering the function of different ion channels. For example, the increased expression of *lncKcna2as* in heart failure reduced $I_{ks}$ and prolonged action potential duration (APD) (*Long et al., 2017*). *LncMALAT1* enhanced arrhythmia susceptibility by suppressing $I_{to}$ and prolonging APD (*Zhu et al., 2018*). In a previous study, we found that *lncCCRR* (cardiac conduction regulatory) interacts with connexin-43 interacting protein 85(CIP85) to promote connexin-43 membrane distribution and improve the impaired cardiac conduction of failing hearts (*Zhang et al., 2018*). However, to date, it remains unknown whether and how lncRNA regulates sodium channel.

*LncDACH1* is an intronic lncRNA located on the first intron of the *DACH1* gene (*Cai et al., 2019*). We previously showed that *lncDach1* impairs cardiac function by promoting the degradation of sarco-endoplasmic reticulum *Atp2a2* (SERCA2a), and exacerbates cardiac ischemia injury by inhibiting Yes-associated protein 1 (YAP1) mediated proliferation of neonatal cardiomyocytes (*Cai et al., 2019*; *Cai et al., 2020*). While analyzing the interacting proteins of *lncDach1* identified by mass spectrometry, we found dystrophin that drove us to hypothesize that *lncDach1* may be a critical regulator of sodium channel Nav1.5.

Therefore, in this study, we explored whether *lncDach1* regulates Nav1.5 by interacting with dystrophin. We found that *lncDach1* inhibited the membrane trafficking of Nav1.5 by binding to dystrophin, which led to reduced sodium current and increased ventricular arrhythmia susceptibility. The study highlights a novel mechanism for the regulation of sodium channel trafficking, and reveals a potential therapeutic target for sodium channel dysfunction-related cardiac arrhythmias.

## Results

### *LncDach1* binds to dystrophin and reduces Nav1.5 membrane distribution in cardiomyocytes of *lncDach1* transgenic mice

We first validated the binding between *lncDach1* and dystrophin. The RNA pulldown plus immuno-blot assay confirmed that *lncDach1* can successfully pulldown dystrophin (*Figure 1A*). These data

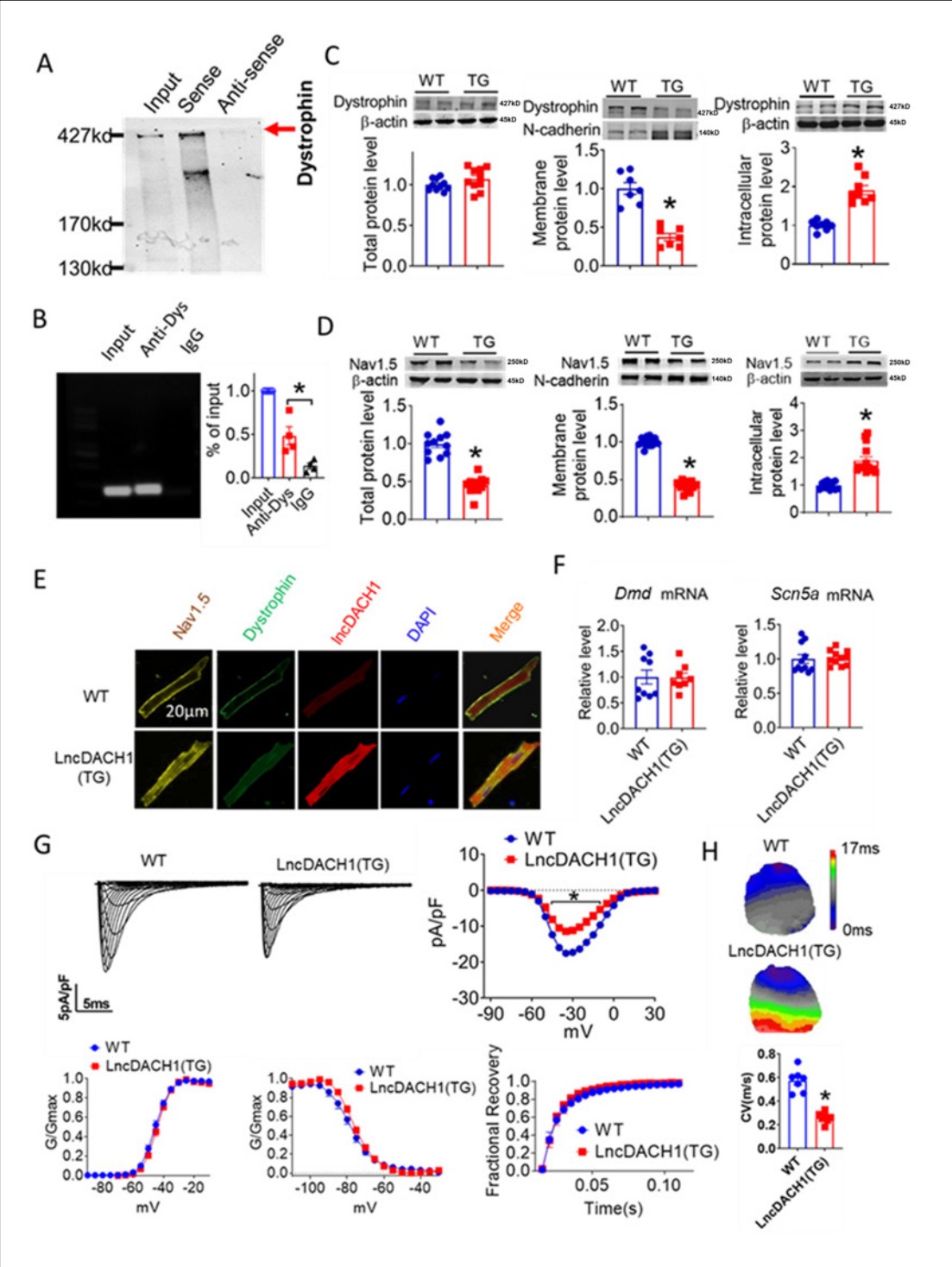

**Figure 1.** Binding of lncRNA-Dachshund homolog 1 (*lncDach1*) to dystrophin and the effects of cardiomyocyte-specific transgenic overexpression of *lncDach1* (*lncDach1*-TG) on the expression and function of sodium channel. (**A**) Blotting of dystrophin pulled-down by *lncDach1*. (**B**) *LncDach1* precipitated by the antibody of dystrophin. N=4. *p<0.05 vs IgG by one-way ANOVA, followed by Tukey's post-hoc analysis. (**C, D**) The total, membrane and intracellular levels of dystrophin and Nav1.5 by Western blot. N-cadherin is the loading control for membrane extracts. N=10–11 for total protein; N=7–14 for membrane protein; N=8–14 for intracellular protein. *p<0.05 vs wild-type (WT) group. p-values were determined by unpaired t-test. (**E**) Distribution of *lncDach1*, dystrophin, and Nav1.5 in isolated cardiomyocytes. (**F**) The mRNA levels of *Dmd* and *Scn5a*. N=8–11. (**G**) Peak $I_{Na}$ currents, I-V curve and kinetics of $I_{Na}$. N=9–21 cells from three mice. *p<0.05 vs WT group. (**H**) Conduction velocity of perfused hearts by optical mapping recordings. N=7. *p<0.05 *vs* WT group. p-values were determined by unpaired t-test.

*Figure 1 continued on next page*

*Figure 1 continued*

The online version of this article includes the following source data and figure supplement(s) for figure 1:

**Source data 1.** Source data for *Figure 1B-D and F-H*.

**Source data 2.** Uncropped and labeled gels for *Figure 1A*.

**Source data 3.** Raw unedited gels for *Figure 1A*.

**Source data 4.** Uncropped and labeled gels for *Figure 1C*.

**Source data 5.** Raw unedited gels for *Figure 1C*.

**Source data 6.** Uncropped and labeled gels for *Figure 1D*.

**Source data 7.** Raw unedited gels for *Figure 1D*.

**Figure supplement 1.** Examination of the potential interaction between lncRNA-Dachshund homolog 1 (*lncDach1*) and Nav1.5.

**Figure supplement 1—source data 1.** Source data for *Figure 1—figure supplement 1B*.

**Figure supplement 1—source data 2.** Uncropped and labeled gels for *Figure 1—figure supplement 1A*.

**Figure supplement 1—source data 3.** Raw unedited gels for *Figure 1—figure supplement 1A*.

**Figure supplement 1—source data 4.** Source data for *Figure 1—figure supplement 1B*.

**Figure supplement 2.** Fluorescence intensity of lncRNA-Dachshund homolog 1 (*lncDach1*), dystrophin, and Nav1.5 in isolated cardiomyocytes of *lncDach1*-TG mice.

**Figure supplement 2—source data 1.** Source data for *Figure 1—figure supplement 2*.

indicated that *lncDach1* does not interact with Nav1.5 directly. Consistently, the RNA immunoprecipitation (RIP) study showed that the antibody for dystrophin precipitated *lncDach1*, while the negative control IgG did not (**Figure 1B**). Conversely, *lncDach1* failed to pulldown Nav1.5, and anti-Nav1.5 did not precipitate *lncDach1* (**Figure 1—figure supplement 1**).

We next explored the influence of *lncDach1* on the cellular distribution of dystrophin. The western blot data showed that the total protein of dystrophin did not change, while the membrane fraction was reduced, and the intracellular fraction was increased in the hearts of *lncDach1*-TG mice than wild-type (WT) controls (**Figure 1C**). Accordingly, the membrane and total protein levels of Nav1.5 were reduced, while intracellular Nav1.5 increased in the hearts of *lncDach1*-TG mice than WT controls (**Figure 1D**). The reduced membrane distribution of dystrophin and Nav1.5 in the cardiomyocytes of *lncDach1*-Tg mice was further confirmed by immunofluorescent staining (**Figure 1E**, **Figure 1—figure supplement 2**). The mRNA levels of *Dystrophin* and *Scn5a* did not change (**Figure 1F**). We then evaluated the functional change of the sodium channel. Consistent with the reduction of membrane Nav1.5, the peak $I_{Na}$ was significantly decreased in the ventricular myocytes of *lncDach1*-TG mice than WT controls, while the kinetics of $I_{Na}$ (activation, inactivation, and recovery) did not change (**Figure 1G**). Meanwhile, the conduction velocity was slower in the hearts of *lncDach1*-TG than WT mice (**Figure 1H**).

We then applied *lncDach1* adenovirus to cultured neonatal cardiomyocytes to confirm the regulation of *lncDach1* on sodium channels in vitro. Infection of adenovirus carrying *lncDach1* significantly upregulated the level of *lncDach1* (**Figure 2A**) and remarkably inhibited peak $I_{Na}$ with no change in kinetics (**Figure 2B and C**). The western blot data showed that the total protein of dystrophin did not change, while the membrane fraction was reduced, and the intracellular fraction increased in neonatal cardiomyocytes with *lncDach1* overexpression (**Figure 2—figure supplement 1**). Consistently, the membrane protein level and total protein level of Nav1.5 were reduced, while intracellular Nav1.5 increased in neonatal cardiomyocytes with *lncDach1* overexpression (**Figure 2—figure supplement 1B**). The membrane distribution of dystrophin and Nav1.5 was remarkably reduced by overexpression of *lncDach1* as indicated by immunofluorescent staining (**Figure 2D**, **Figure 2—figure supplement 2**). The mRNA levels of *Dmd* and *Scn5a* were not altered by *lncDach1* overexpression (**Figure 2E**).

## Inhibition of *lncDach1* in cardiomyocytes increased membrane Nav1.5 distribution

We then employed *lncDach1* cardiomyocyte conditional knockout (*lncDach1*-cKO) mice to examine the regulation of *lncDach1* on dystrophin and Nav1.5. The western blot data showed that membrane distribution of dystrophin was increased in the hearts of *lncDach1*-cKO mice than WT controls, while

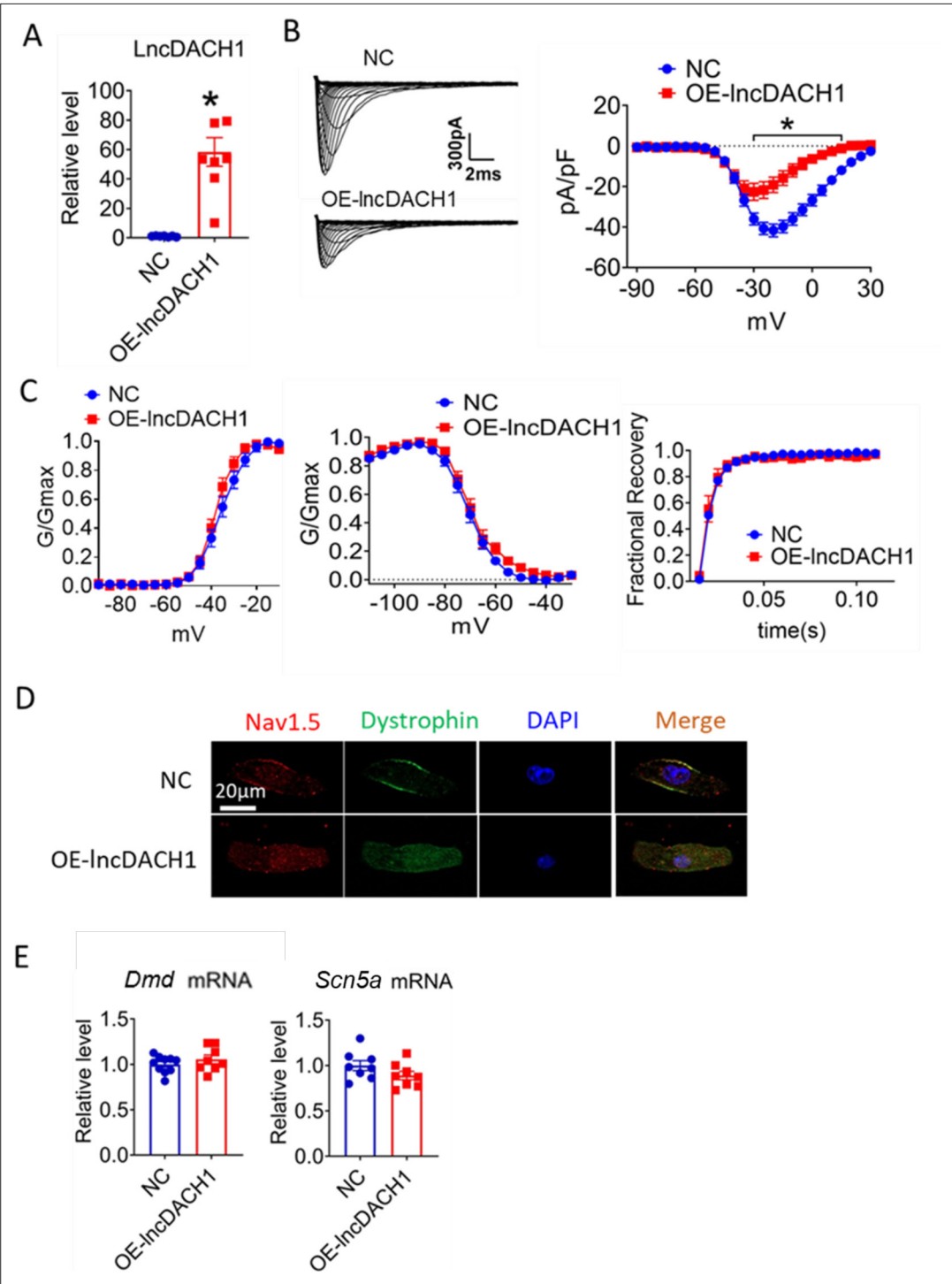

**Figure 2.** Effects of lncRNA-Dachshund homolog 1 (*lncDach1*) overexpression on sodium channel expression and function in cultured neonatal cardiomyocytes. (**A**) Verification of the expression of *lncDach1* after transfection of adenovirus carrying *lncDach1*. N=7–8 from three independent cultures. *p<0.05 vs NC (negative control, empty plasmid). p-values were determined by unpaired t-test. (**B, C**) Peak $I_{Na}$ currents, I-V curves and kinetics of $I_{Na}$. N=7–25 cells from three independent cultures. *p<0.05 vs NC. (**D**) Distribution of Nav1.5 and dystrophin by immunofluorescent staining. (**E**) The mRNA levels of *Dmd* and *Scn5a*. N=8–10 from hree independent cultures.

The online version of this article includes the following source data and figure supplement(s) for figure 2:

**Source data 1.** Source data for *Figure 2A–C and E*.

*Figure 2 continued on next page*

*Figure 2 continued*

**Figure supplement 1.** Overexpression of *lncDach1*(oe-*lncDach1*) in cultured neonatal cardiomyocytes decreased membrane dystrophin and Nav1.5 expression.

**Figure supplement 1—source data 1.** Source data for *Figure 2—figure supplement 1A–B*.

**Figure supplement 1—source data 2.** Uncropped and labeled gels for *Figure 2—figure supplement 1A*.

**Figure supplement 1—source data 3.** Raw unedited gels for *Figure 2—figure supplement 1A*.

**Figure supplement 1—source data 4.** Uncropped and labeled gels for *Figure 2—figure supplement 1B*.

**Figure supplement 1—source data 5.** Raw unedited gels for *Figure 2—figure supplement 1B*.

**Figure supplement 2.** Fluorescence intensity of dystrophin and Nav1.5 in cultured neonatal cardiomyocyte overexpressing lncRNA-Dachshund homolog 1 (*lncDach1*).

**Figure supplement 2—source data 1.** Source data for *Figure 2—figure supplement 2*.

the total dystrophin protein and *Dystrophin* mRNA did not change (*Figure 3A*). Consistently, the membrane and total level of Nav1.5 was increased in the hearts of *lncDach1*-cKO mice than WT controls, with no change in *Scn5a* mRNA (*Figure 3B*). The change in dystrophin and Nav1.5 membrane distribution was further validated by immunofluorescent staining (*Figure 3C*, *Figure 3—figure supplement 1*). Meanwhile, the peak $I_{Na}$ was larger in cardiomyocytes of *lncDach1*-cKO mice than WT controls, while the kinetics of $I_{Na}$ (activation, inactivation, and recovery) did not change (*Figure 3D*). Consistent with the increase of peak $I_{Na}$, the conduction velocity in the hearts of *lncDach1*-cKO mice was faster than WT controls (*Figure 3E*).

We further confirmed the effects of *lncDach1* knockdown with its shRNA on sodium channels in cultured neonatal cardiomyocytes in vitro. Infection of adenovirus carrying shRNA for *lncDach1* significantly reduced the level of *lncDach1* (*Figure 4A*). The patch-clamp recordings showed that the knockdown of *lncDach1* significantly increased the current density of peak $I_{Na}$ with no change in channel kinetics (*Figure 4B and C*). The western blot data showed that the total protein of dystrophin did not change, while the membrane fraction was increased, and the intracellular fraction was reduced after *lncDach1* knockdown (*Figure 4—figure supplement 1A*). Consistently, the membrane and total protein levels of Nav1.5 were increased, while intracellular Nav1.5 reduced after *lncDach1* knockdown (*Figure 4—figure supplement 1B*). Membrane distribution of dystrophin and Nav1.5 were both increased after the knockdown of *lncDach1* as indicated by immunofluorescent staining (*Figure 4D*, *Figure 4—figure supplement 2*). The mRNA levels of *Dmd* and *Scn5a* were not altered by *lncDach1* knockdown (*Figure 4E*).

In addition, neither knockout nor transgenic overexpression of *lncDach1* changed the expression of *Dach1* and its neighbor genes (*Klhl1, Bora, Mzt1, Dis3*) (*Figure 4—figure supplement 3*).

## Transgenic overexpression of *lncDach1* is arrhythmogenic in mice

We next evaluated whether the inhibition of Nav1.5 by *lncDach1* is arrhythmogenic in *lncDach1*-TG mice. Electrical pacing technique was employed to evaluate the arrhythmia susceptibility of intact hearts in vivo and isolated hearts ex vivo. The in vivo study showed that programmed pacing induced more ventricular arrhythmia in *lncDach1*-TG mice that WT controls. Both induction rate and episodes of ventricular arrhythmia were higher in *lncDach1*-TG mice (*Figure 5A*). Consistent with the results of in vivo., the ex vivo electrical pacing study demonstrated that ventricular arrhythmia was more frequently occurred in *lncDach1*-TG mice (*Figure 5B*). The optical mapping study revealed that there are more breaking points in the perfused heart of *lncDach1*-TG mice than WT controls (*Figure 5C and D*). Conversely, no ventricular arrhythmia was induced in the hearts of *lncDach1*-cKO mice (*Figure 5E*).

## The conservative fragment of *lncDach1* reduced peak sodium current and promoted ventricular arrhythmia

The sequence blasting data showed that the fragment of *lncDach1* from 835 to 2085 nts is conservative between humans and mouse. We then cut *lncDach1* into different fragments (*Figure 6A*) to evaluate the sequence that is responsible for the binding with dystrophin. The data showed that only fragments containing the conserved sequence, fragments a and b, can pulldown dystrophin, indicating that the conserved sequence may be the functional region of *lncDach1* (*Figure 6A*).

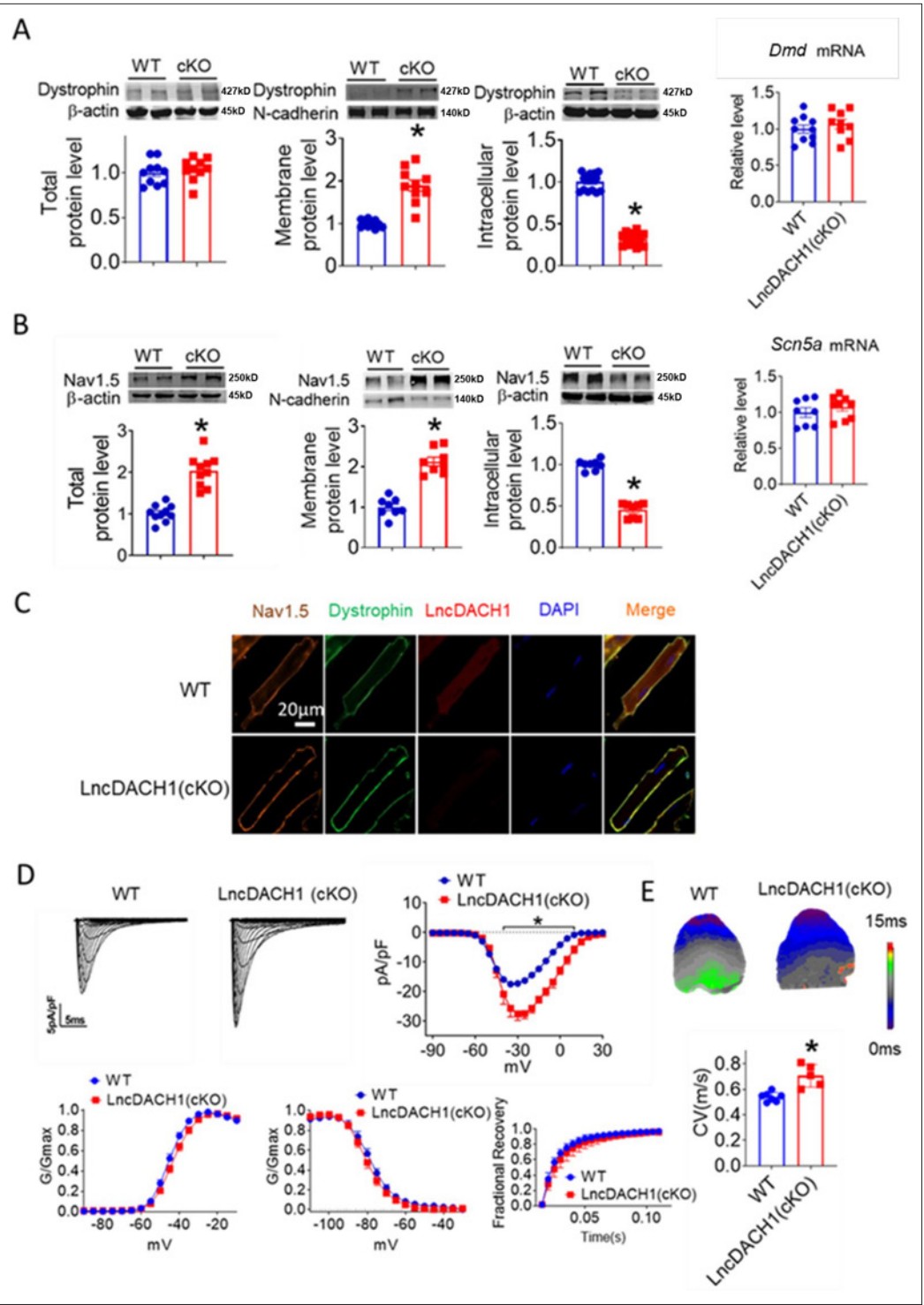

**Figure 3.** Conditional knockout of negative control (*lncDach1*) (*lncDach1*-cKO) in cardiomyocytes increased peak sodium current, membrane Nav1.5 expression. (**A**) The total, membrane and intracellular levels of dystrophin by Western blot and *Dmd* mRNA by qRT-PCR. N-cadherin is the loading control for membrane extracts. N=10 for total protein; N=10 for membrane protein; N=15 for intracellular protein. *p<0.05 vs ild-type (WT) group. p-values were determined by unpaired t-test. (**B**) The total, membrane and intracellular levels of Nav1.5 by Western blot and *Scn5a* mRNA by qRT-PCR. N-cadherin is the loading control for membrane extracts. N=10 for total protein; N=8 for membrane protein; N=8 for intracellular protein. *p<0.05 vs WT group. p-values were determined by unpaired t-test. (**C**) Distribution of *lncDach1*, dystrophin, and Nav1.5 in isolated cardiomyocytes. (**D**) Peak $I_{Na}$ currents, I-V

*Figure 3 continued on next page*

*Figure 3 continued*

curve, and kinetics of $I_{Na}$. N=7–21 cells; N=3 mice of WT; N=4 mice of *LncDach1* (cKO). *p<0.05 vs WT group. (**E**) Conduction velocity of perfused hearts by optical mapping recordings. N=7 and 5. *p<0.05 vs WT group. p-values were determined by unpaired t-test.

The online version of this article includes the following source data and figure supplement(s) for figure 3:

**Source data 1.** Source data for *Figure 3A-B and D-E*.

**Source data 2.** Uncropped and labeled gels for *Figure 3A*.

**Source data 3.** Raw unedited gels for *Figure 3A*.

**Source data 4.** Uncropped and labeled gels for *Figure 3B*.

**Source data 5.** Raw unedited gels for *Figure 3B*.

**Figure supplement 1.** Fluorescence intensity of lncRNA-Dachshund homolog 1 (*lncDach1*), dystrophin, and Nav1.5 in isolated cardiomyocytes of *lncDach1*-cKO mice.

**Figure supplement 1—source data 1.** Source data for *Figure 3—figure supplement 1*.

We then examined the influence of the conserved sequence from 835 to 2085 nts (conserved fragment of *lncDach1*, cF- *lncDach1*) on the cardiac sodium channel. The adenovirus carrying cF- *lncDach1* was constructed and administered to mice. The successful overexpression of cF- *lncDach1* was validated by qRT-PCR (*Figure 6B*). Administration of cF- *lncDach1* reduced the membrane distribution, and increased intracellular expression of both dystrophin and Nav1.5 as indicated by western blot and immunofluorescent data (*Figure 6C, D*, *Figure 6—figure supplement 1*). The mRNA levels of *Dmd* and *Scn5a* were not affected by cF- *lncDach1* (*Figure 6—figure supplement 2A*). Overexpression of cF- *lncDach1* reduced peak $I_{Na}$ currents (*Figure 6E*), and produced no influence on channel kinetics (*Figure 6—figure supplement 2B*). The optical mapping data showed that administration of cF- *lncDach1* reduced conduction velocity and increased breakpoints of ventricular arrhythmias (*Figure 6F and G*). The induction rate and episodes of ventricular tachycardia (VT) were higher in the cF- *lncDach1* group than in controls (*Figure 6H*).

In cultured neonatal cardiomyocytes, overexpression of cF- *lncDach1* reduced peak $I_{Na}$ with no change in kinetics, inhibited membrane distribution of dystrophin and Nav1.5, and produced no influence on the mRNA levels of *Dmd* and *Scn5a* (*Figure 6—figure supplements 3 and 4*). The western blot data showed that the total protein of dystrophin did not change, while the membrane fraction was reduced, and the intracellular fraction increased after overexpression of cF- *lncDach1* (*Figure 6—figure supplement 5A*). Consistently, the membrane and total protein levels of Nav1.5 were reduced, while intracellular Nav1.5 increased after overexpression of cF- *lncDach1* (*Figure 6—figure supplement 5B*).

We also blasted the *lncDACH1* in different species and found that *lncDACH1* is conserved among sheep, pigs, dogs, rats, humans and mouse (*Figure 6—figure supplement 6*).

## Activation of dystrophin transcription by the dCas9-SAM system prevented the reduction of sodium current in *lncDach1* transgenic mice

As *lncDach1* reduced Nav1.5 membrane targeting by interacting with dystrophin, we reasoned that overexpression of dystrophin may rescue the inhibition of Nav1.5 by *lncDach1*. To test this notion, we constructed the AAV9 virus carrying dCas9-SAM system that can activate *Dmd* transcription (AAV9-Dys-Act) to perform rescuing experiments on *lncDach1*-TG mice (*Figure 7—figure supplement 1*). Tail vein injection of the AAV9-Dys-Act virus significantly increased the mRNA level of *Dystrophin* in the hearts of both WT and *lncDach1*-TG mice (*Figure 7A*). The western blot data showed that overexpression of dystrophin with AAV9-Dys-Act virus increased both total and membrane protein expression of dystrophin, and rescued the reduction of dystrophin expression in *lncDach1*-TG mice (*Figure 7A*). The mRNA level of *Scn5a* was not influenced by AAV9-Dys-Act virus (*Figure 7B*). Strikingly, AAV9-Dys-Act virus administration restored the total and membrane expression of Nav1.5 in *lncDach1*-TG mice (*Figure 7B*). In addition, AAV9-Dys-Act virus injection rescued the reduction of peak $I_{Na}$ current in *lncDach1*-TG mice (*Figure 7C*). The kinetics of $I_{Na}$ current did not change among groups (*Figure 7C*). Activation of dystrophin transcription with AAV9-Dys-Act virus restored the conduction velocity in *lncDach1*-TG mice (*Figure 7D*). Both in vivo and ex vivo data indicated that activation of dystrophin

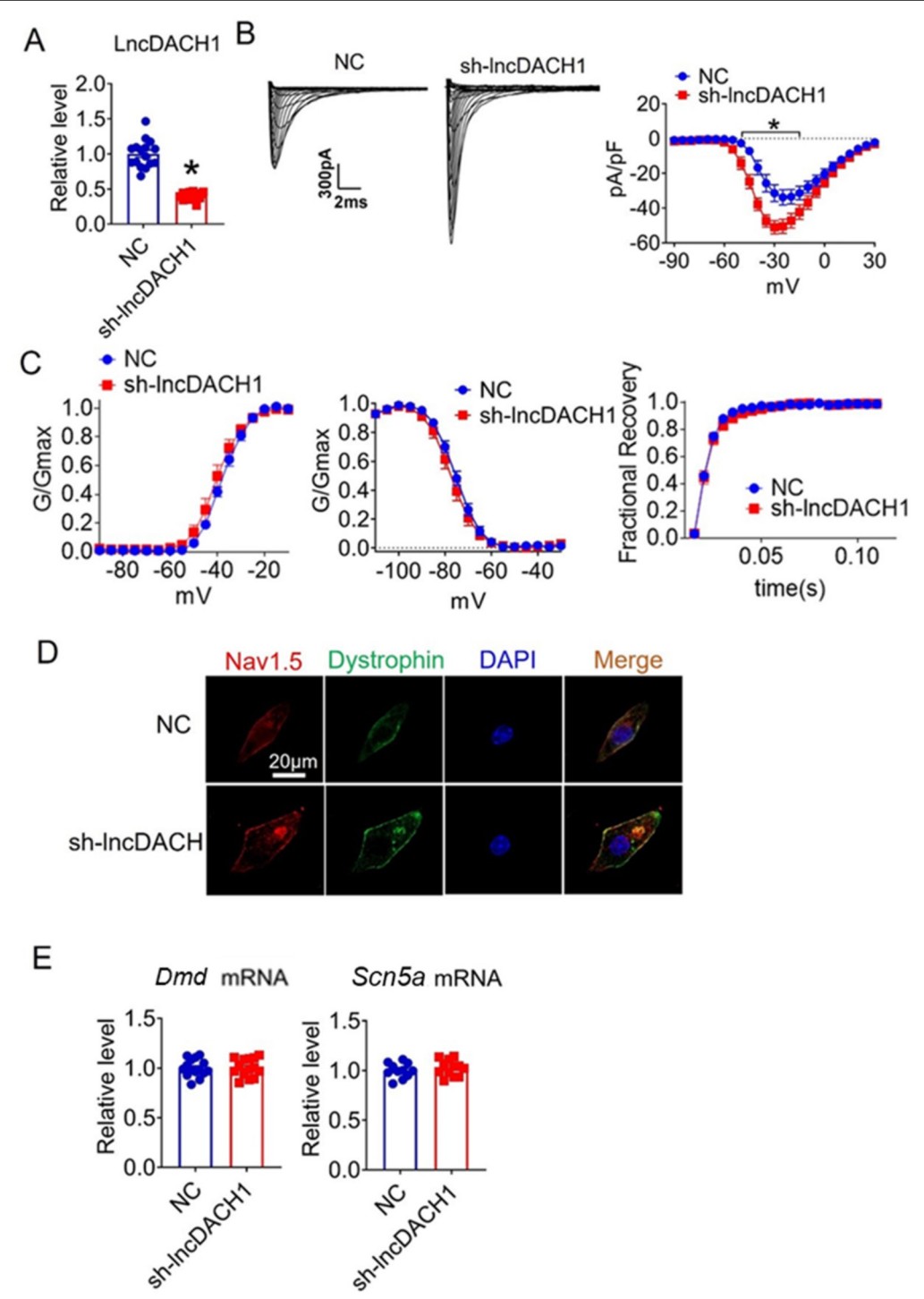

**Figure 4.** Effects of lncRNA-Dachshund homolog 1 (*lncDach1*) knockdown on sodium channel expression and function in cultured neonatal cardiomyocytes. (**A**) Verification of the expression of *lncDach1* after infection of adenovirus carrying *lncDach1* shRNA. N=15 from three independent cultures. *p<0.05 vs NC (negative control, empty plasmids). p-values were determined by unpaired t-test. (**B, C**) Peak $I_{Na}$ currents, I-V curve and kinetics of $I_{Na}$. N=8–15 from three independent cultures. *p<0.05 vs NC. (**D**) Distribution of Nav1.5 and dystrophin by immunofluorescent staining. (**E**) The mRNA levels of *Dmd* and *Scn5a*. N=11–15 from three independent cultures.

The online version of this article includes the following source data and figure supplement(s) for figure 4:

**Source data 1.** Source data for *Figure 4A–C and E*.

*Figure 4 continued on next page*

*Figure 4 continued*

**Figure supplement 1.** Effects of lncRNA-Dachshund homolog 1 (*lncDach1*) knockdown with shRNA on the protein expression of dystrophin and Nav1.5 expression in cultured neonatal cardiomyocytes.

**Figure supplement 1—source data 1.** Source data for *Figure 4—figure supplement 1A–B*.

**Figure supplement 1—source data 2.** Uncropped and labeled gels for *Figure 4—figure supplement 1A*.

**Figure supplement 1—source data 3.** Raw unedited gels for *Figure 4—figure supplement 1A*.

**Figure supplement 1—source data 4.** Uncropped and labeled gels for *Figure 4—figure supplement 1B*.

**Figure supplement 1—source data 5.** Raw unedited gels for *Figure 4—figure supplement 1B*.

**Figure supplement 2.** Fluorescence intensity of dystrophin and Nav1.5 in cultured neonatal cardiomyocytes after knocking down of lncRNA-Dachshund homolog 1 (*lncDach1*).

**Figure supplement 2—source data 1.** Source data for *Figure 4—figure supplement 2*.

**Figure supplement 3.** The mRNA expression of *Klhl1*, *Bora*, *Mzt1*, *Dis3*, and *Dach1* in the cardiac tissue of lncRNA-Dachshund homolog 1 (*lncDach1*) knockout (**A**) and transgenic overexpression (**B**) mice.

**Figure supplement 3—source data 1.** Source data for *Figure 4—figure supplement 3*.

transcription reduced the susceptibility to ventricular arrhythmia of *lncDach1*-TG mice (*Figure 7E and F*).

## Hadhb binds to *lncDach1* and promotes its decay

Reduced Nav1.5 expression and reduction of peak $I_{Na}$ in heart failure have been reported by multiple studies (*Baba et al., 2005*; *Valdivia et al., 2005*; *Xi et al., 2009*; *Dybkova et al., 2018*). We therefore evaluated the contribution of *lncDach1* on sodium channel remodeling in transaortic constriction (TAC) induced heart failure model in mice. We found that *lncDach1* was increased in failing hearts than sham controls (*Figure 8A*). Although *lncDach1* was upregulated in failing hearts, the mRNA of its host gene *Dach1* did not change (*Figure 8B*). This finding excluded the transcription-related mechanism of *lncDach1* upregulation during heart failure. By analyzing the RNA Pulldown plus Mass Spectrometry data, we identified three potential interacting proteins of *lncDach1* that have been shown to regulate RNA stability. They are *Anp32a* (acidic leucine-rich nuclear phosphoprotein 32 A), *Eif4a1* (eukaryotic initiation factor 4A1), and *Hadhb* (hydroxyacyl-CoA dehydrogenase subunit beta). We therefore speculated that it may be the change of RNA stability that renders to the expression change of *lncDach1*. We then tested whether these proteins can affect *lncDach1* level by knocking down their expression with siRNA. The data showed that the knockdown of *Hadhb* increased the expression of *lncDach1*, while knockdown of *Anp32a* and *Eif4a1* produced no influence (*Figure 8C*). The influence of *Hadhb* on *lncDach1* stability was further validated by the fact that the knockdown of *Hadhb* increased the decaying half-life of *lncDach1* (*Figure 8D*). Furthermore, the sense sequence of *lncDach1* successfully pulled down hadhb, and the antibody of hadhb precipitated *lncDach1* (*Figure 8E*). Additionally, the protein level of hadhb was reduced in mouse failing hearts (*Figure 8F*), which is inversely correlated to the upregulation of *lncDach1*. The siRNA for *Hadhb* reduced the expression of Nav1.5 (*Figure 8G*). These data indicated that *Hadhb* is an upstream regulator of *lncDach1* which determines the stability of *lncDach1*.

We lastly evaluated the human conserved sequence of *lncDACH1* (hcF- *lncDACH1*) on Nav1.5 distribution of human iPS-induced cardiomyocytes. We found that overexpression of hcF- *lncDACH1* reduced the membrane distribution of Nav1.5 (*Figure 8H*, *Figure 8—figure supplement 1*). To confirm the arrhythmogenic effects of hcF- *lncDACH1* in the hiPSC-CM model, we overexpressed hcF- *lncDACH1*. We found that overexpression of hcF- *lncDACH1* significantly inhibited sodium current and the Vmax of APD upstroke (*Figure 8—figure supplement 2*). These data indicate the arrhythmogenic effects of hcF- *lncDACH1*, and imply that knockdown of hcF- *lncDACH1* may reduce the susceptibility of arrhythmia.

## Discussion

In this study, we discovered that *lncDACH1* is a critical regulator of the sodium channel in the heart. *lncDACH1* binds to dystrophin and thus inhibits membrane trafficking of Nav1.5, which leads to the

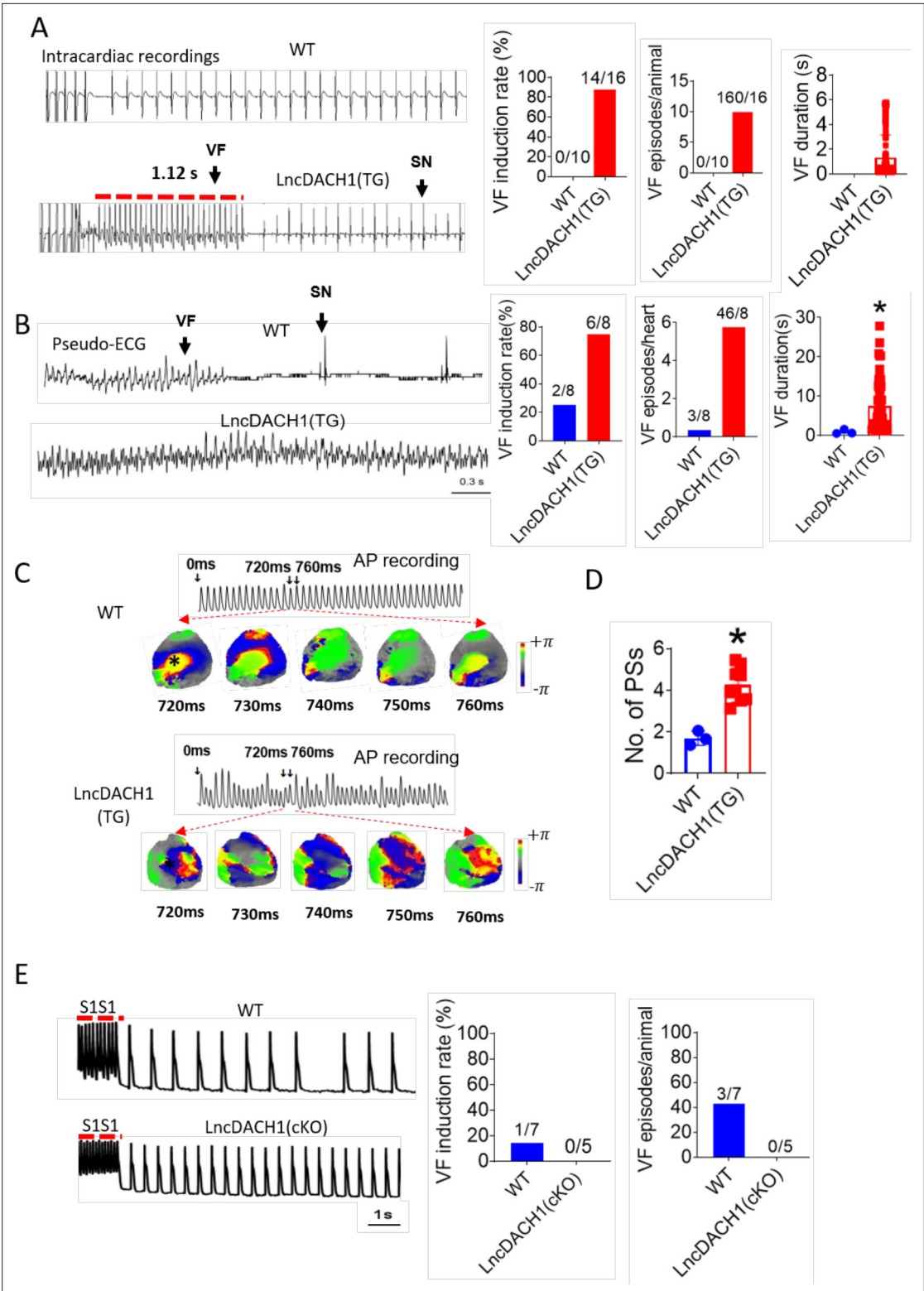

**Figure 5.** Increased arrhythmia susceptibility in lncRNA-Dachshund homolog 1 (*lncDach1*)-TG mice. (**A**) Ventricular fibrillation (VF) induced by S1S2 pacing in intact mice. The red lines in the ECG traces indicate VF duration. N=10–16. SN: sinus rhythm (**B**) VF induced by S1S1 pacing of perfused hearts. (**C**) Break points during VT of WT and *lncDach1*-TG mice by optical mapping. Consecutive phase maps sampled at 10 ms interval during VF from WT and TG mice. Phase singularities (wavebreaks) are indicated by phase maps. Upper panels showed a corresponding optical recording of VF at

*Figure 5 continued on next page*

*Figure 5 continued*

asterisk site. N=8. (**D**) The number of phase singularities and dominant frequency of WT and TG mice. (**E**) VF induced by S1S1 pacing in perfused hearts. N=5–7 mice.

The online version of this article includes the following source data for figure 5:

**Source data 1.** Source data for *Figure 5A-B and D-E*.

reduction of peak sodium current and impairment of cardiac conduction. Therefore, upregulation of *lncDACH1* increased the susceptibility to ventricular arrhythmia (*Figure 8I*).

LncRNAs have been established to be critical regulators of various biological processes (*Yao et al., 2019*). The action modes of lncRNAs are complex. One major mechanism for them to exert their biological function in the intracellular is to interact with such as proteins, miRNAs, and mRNAs to alter protein translation, enzyme activity, protein degradation, etc (*Ulitsky and Bartel, 2013*). For instance, lncRNA- *CCRR* was shown to inhibit the endocytic trafficking of connexin-43 by binding to CIP85 (*Zhang et al., 2018*). *LncDACH1* mainly distributes in the intracellular, and can bind to SERCA2a to promote its ubiquitination and degradation (*Cai et al., 2019*). *lncDACH1* can also bind to protein phosphatase 1 catalytic subunit alpha (PP1A) to inhibit its dephosphorylation activity on YAP1, leading to the intracellular sequestration of YAP1 (*Cai et al., 2020*). The unraveling of the molecular mechanism of Nav1.5 is critical for the insightful understanding of sodium channel function under physiological and pathological conditions. Several interacting proteins have been demonstrated to determine the membrane distribution of Nav1.5 and sodium channel function (*Abriel et al., 2015*). Dystrophin is a well-characterized Nav1.5 partner protein. It indirectly interacts with Nav1.5 via syntrophin, which binds with the C-terminus of dystrophin and with the SIV motif on the C-terminus of Nav1.5 (*Gavillet et al., 2006*; *Shy et al., 2014*). In this study, we found that *lncDACH1* binds to dystrophin and leads to the impairment of Nav1.5 trafficking and reduced membrane distribution.

Although the membrane distribution of both dystrophin and Nav1.5 was inhibited by *lncDach1*, the total protein level of dystrophin was not affected, while Nav1.5 was reduced. The mechanism for the differential influence of *lncDach1* on total protein levels of dystrophin and Nav1.5 is unclear. One explanation may be that Nav1.5 is a membrane-channel protein. If they failed to target on the plasma membrane, they may be regarded as unnecessary proteins and undergo the process of protein degradation. The E1-E3 enzymes in the ubiquitination systems have been shown to regulate the degradation of Nav1.5, which includes E1 enzyme UBE1 (Ubiquitin-activating Enzyme1), UBA6 (Ubiquitin-like modifier-activating enzyme 6), E2 enzyme, UBC9 (Ubiquitin-Conjugating Enzyme 9), and E3 ligase Nedd4-2 (neuronal precursor cell expressed developmentally downregulated 4–2) (*Tang et al., 2019*; *van Bemmelen et al., 2004*; *Hu et al., 2020*). LITAF (lipopolysaccharide-induced tumor necrosis factor-alpha factor), a protein-encoding a regulator of endosomal trafficking, was shown to reduce surface Nav1.5 by promoting degradation of NEDD4-2 (*Turan et al., 2020*). Therefore, intracellular Nav1.5 that failed to target on plasma membrane may be quickly distinguished and then degraded by these ubiquitination enzymes (*Figure 8—figure supplement 3*).

The dysfunction of the sodium channel is associated with various arrhythmias. Reduced peak $I_{Na}$ due to *Scn5a* loss-of-function mutation can cause a series of arrhythmias such as atrial fibrillation, Brugada syndrome, long QT syndrome, sudden cardiac death, and ventricular tachycardia etc (*Han et al., 2018*; *Savio-Galimberti et al., 2018*). Consistently, we found that accompanied by the reduction of peak $I_{Na}$, overexpression of *lncDach1* reduced ventricular conduction velocity and increased the susceptibility to ventricular arrhythmia in mice. The increased peak $I_{Na}$ due to *Scn5a* gain-of-function mutation is associated with arrhythmias such as atrial fibrillation, long QT syndrome; polymorphic ventricular complexes, and ventricular arrhythmia (*Han et al., 2018*; *Savio-Galimberti et al., 2018*). In this study, although the peak $I_{Na}$ increased in *lncDach1*-cKO mice, the susceptibility to arrhythmia did not increase. One difference of our data with *Scn5a* gain-of-function mutation is that the kinetics of peak $I_{Na}$ often change during mutation, while it is not the case of *lncDach1* knockout. One possible explanation may be that *lncDach1* does not alter Nav1.5 gating, and the late Na current may not be enhanced to the same effect as observed with LQT gain-of-function Nav1.5 mutations, in which APD prolongation is attributed to gating defects that increase late Na current.

Sodium channel remodeling occurs commonly in cardiac diseases, especially heart failure. Despite of some discrepancies, the main observations are that the peak $I_{Na}$, *Scn5a* mRNA, and Nav1.5 protein are reduced during heart failure in both human patients and animal models (*Baba et al., 2005*;

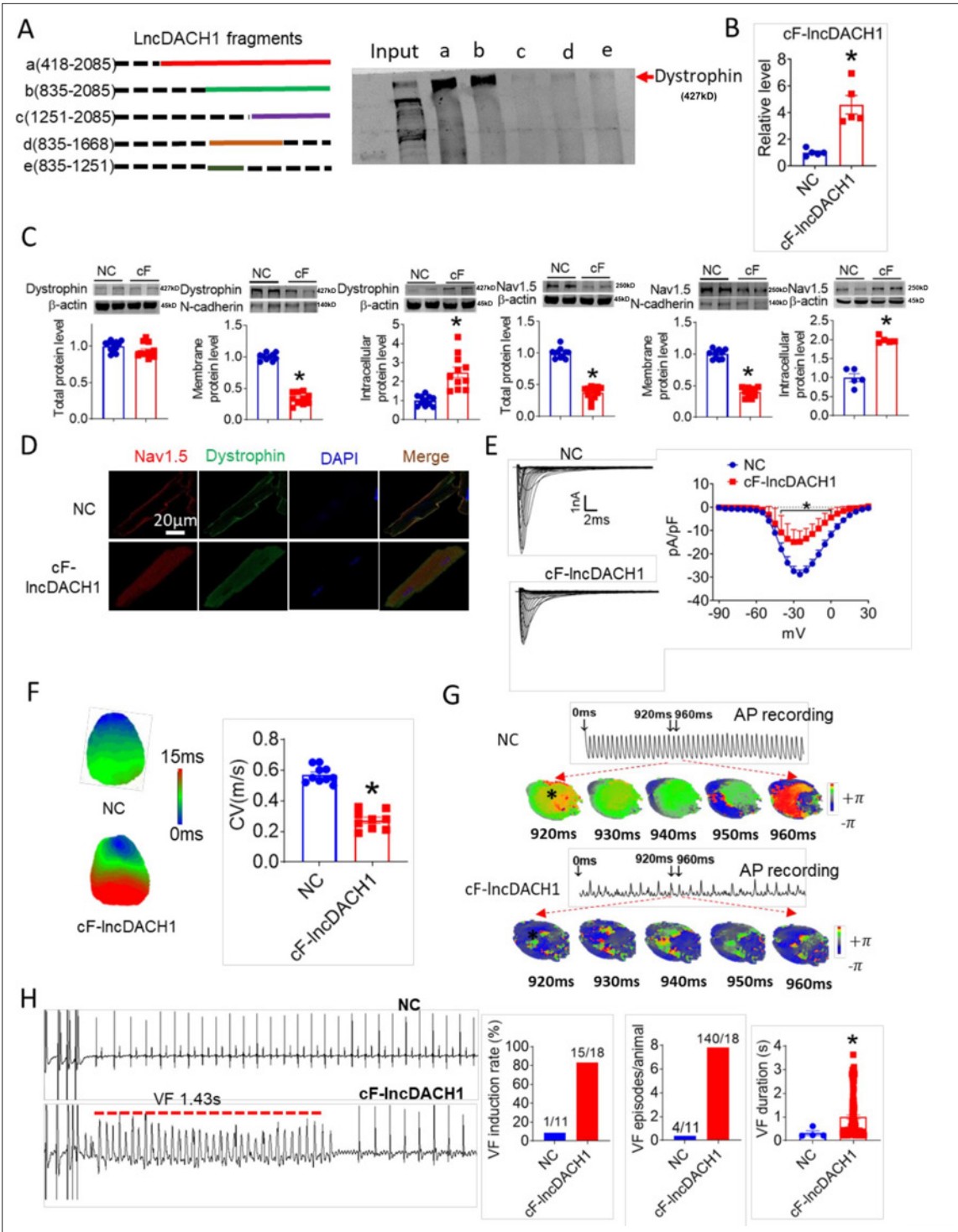

**Figure 6.** The conserved fragment of lncRNA-Dachshund homolog 1 (*lncDach1*) (cF-*lncDach1*) inhibited sodium channel function in mice. (**A**) Pulldown of dystrophin by fragments of *lncDach1* as indicated. (**B**) Verification of the expression of cF-*lncDach1* after injection of adeno-virus carrying cF-*lncDach1*. N=5. *p<0.05 vs NC (negative control, Adeno-virus carrying empty plasmid). P-values were determined by unpaired t-test. (**C**) The total, membrane and intracellular levels of dystrophin and Nav1.5 by Western blot. N-cadherin is the loading control for membrane extracts. N=11–12 for total protein; N=10 for membrane protein; N=5–11 for intracellular protein. *p<0.05 vs NC group. P-values were determined by unpaired t-test. (**D**) Distribution of dystrophin and Nav1.5 in isolated cardiomyocytes. (**E**) Representative traces and I-V curve of peak $I_{Na}$ currents. N=20–29 cells; N=3 mice of NC; N=4 mice of cF-*lncDach1*. *p<0.05 *vs* NC group. (**F**) Conduction velocity of perfused hearts by optical mapping recordings. N=9–10. *p<0.05 vs NC group. p-values were determined by unpaired t-test. (**G**) Break points during ventricular tachycardia (VT) by optical mapping. (**H**) Ventricular

*Figure 6 continued on next page*

*Figure 6 continued*

fibrillation (VF) induced by S1S2 pacing in intact mice. The induction rate, average episodes and duration of VF were determined from ECG recordings. The red lines in the ECG traces indicate VF duration. N=11 for NC; N=18 for cF-*lncDach1*. *p<0.05 vs NC group.

The online version of this article includes the following source data and figure supplement(s) for figure 6:

**Source data 1.** Source data for *Figure 6B–C, E–F and H*.

**Source data 2.** Uncropped and labeled gels for *Figure 6A*.

**Source data 3.** Raw unedited gels for *Figure 6A*.

**Source data 4.** Uncropped and labeled gels for *Figure 6C*.

**Source data 5.** Raw unedited gels for *Figure 6C*.

**Figure supplement 1.** Fluorescence intensity of dystrophin and Nav1.5 in isolated cardiomyocytes overexpressing cF-*lncDach1*.

**Figure supplement 1—source data 1.** Source data for *Figure 6—figure supplement 1*.

**Figure supplement 2.** The conserved fragment of *lncDach1*(cF-*lncDach1*) inhibited sodium channel function in mice.

**Figure supplement 2—source data 1.** Source data for *Figure 6—figure supplement 2*.

**Figure supplement 3.** Effects of cF-*lncDach1* overexpression on sodium channel expression and function in cultured neonatal cardiomyocytes.

**Figure supplement 3—source data 1.** Source data for *Figure 6—figure supplement 3A–B and D*.

**Figure supplement 4.** Fluorescence intensity of dystrophin and Nav1.5 in cultured neonatal cardiomyocytes overexpressing cF-*lncDach1*.

**Figure supplement 4—source data 1.** Source data for *Figure 6—figure supplement 4*.

**Figure supplement 5.** Overexpression of cF-*lncDach1*(oe-cF-*lncDach1*) in cultured neonatal cardiomyocytes decreased membrane dystrophin and Nav1.5 expression.

**Figure supplement 5—source data 1.** Source data for *Figure 6—figure supplement 5A–B*.

**Figure supplement 5—source data 2.** Uncropped and labeled gels for *Figure 6—figure supplement 5A*.

**Figure supplement 5—source data 3.** Raw unedited gels for *Figure 6—figure supplement 5A*.

**Figure supplement 5—source data 4.** Uncropped and labeled gels for *Figure 6—figure supplement 5B*.

**Figure supplement 5—source data 5.** Raw unedited gels for *Figure 6—figure supplement 5B*.

**Figure supplement 6.** Conservation of *lncDACH1*.

**Figure supplement 6—source data 1.** Source data for *Figure 6—figure supplement 6B*.

*Valdivia et al., 2005*; *Xi et al., 2009*; *Dybkova et al., 2018*). In this study, we found that *lncDach1* was increased during heart failure, indicating that it may contribute to sodium channel remodeling and arrhythmogenesis during heart failure by interfering with the action of dystrophin. Interestingly, we found that activation of dystrophin with dCas9-SAM system restored the membrane distribution of Nav1.5 in cardiomyocytes of *lncDach1*-TG mice, which implies its potential in counteracting sodium channel remodeling of patients with heart failure.

The low sequence conservation of lncRNAs among species is a critical issue that limits the extrapolation of data from animal studies to human beings (*Johnsson et al., 2014*; *Tsagakis et al., 2020*). In this study, we found that the conserved fragment of *lncDach1* exhibits the same effect as *lncDach1* on Nav1.5 trafficking and arrhythmogenesis. Moreover, the human conservative homologous fragment of *lncDACH1* can inhibit the membrane distribution of Nav1.5 in cardiomyocytes derived from induced pluripotent stem cells (iPS-CMs). These findings hint at the clinical relevance of *lncDACH1*, which holds the potential to become a therapeutic target for treating sodium channel remodeling in clinic.

In conclusion, *lncDACH1* is a novel regulator of sodium channels, which suppresses the membrane trafficking of Nav1.5 by disturbing the function of dystrophin. The current work enriched our understanding of the biology of sodium channel trafficking and function, and indicated that lncRNAs possess the potential to become therapeutic targets for ventricular arrhythmias.

## Materials and methods
### Animals

Neonatal (within 3 d post-born) and adult C57BL/6 mice (8–10 wk old) were provided by the animal center at the Second Affiliated Hospital of Harbin Medical University. Use of animals was approved by the IRB of the College of Pharmacy, Harbin Medical University, and conformed to the Guide for the

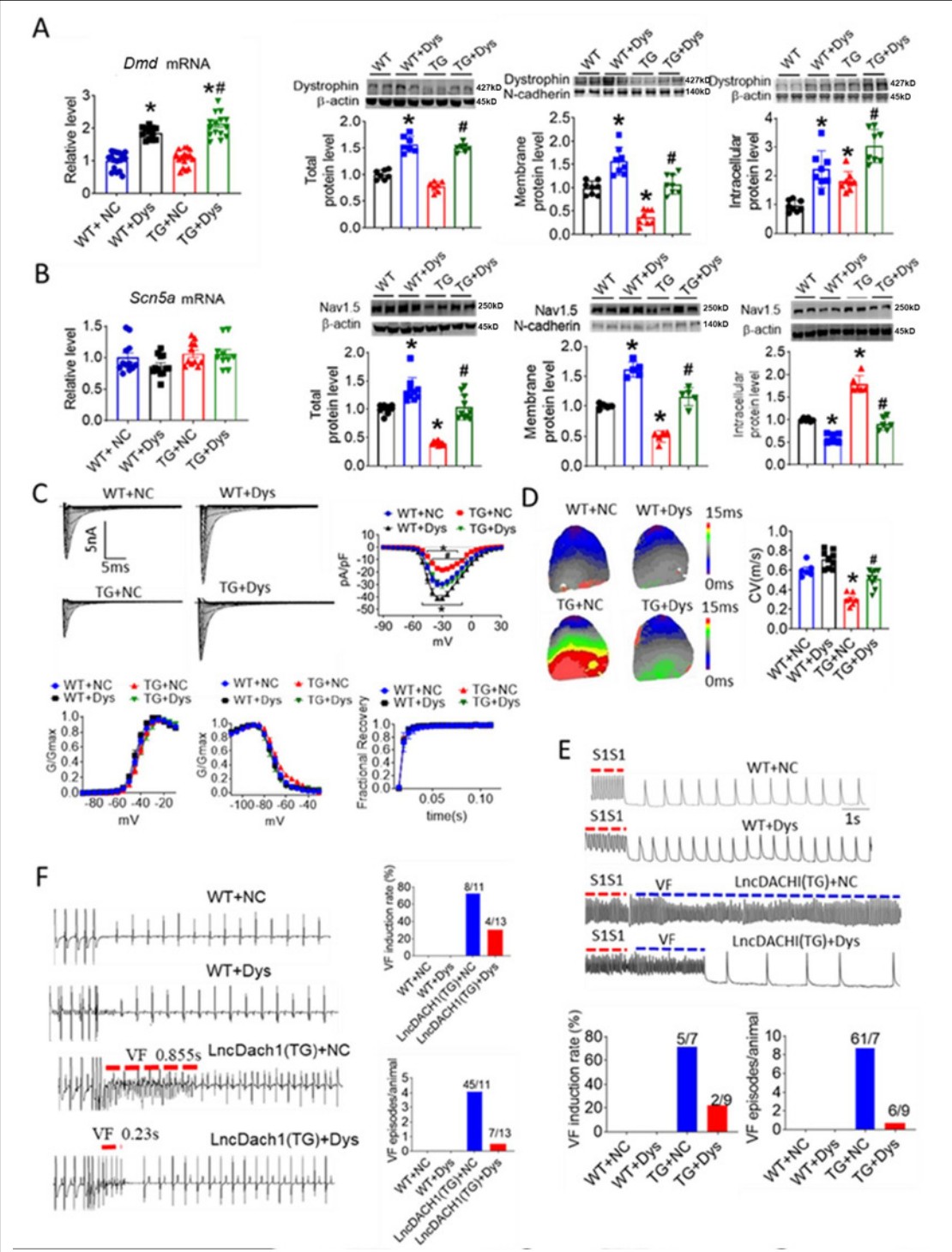

**Figure 7.** Activation of dystrophin transcription by adeno-associated virus 9 (AAV9) virus carrying dCas9-SAM system (AAV9-Dys-Act) rescued the remodeling of sodium channel in *lncDach1*-TG mice. (**A**) The mRNA level of *Dmd* by real-time PCR (N=12–17) and the total, membrane and intracellular protein levels of dystrophin by western blot. N-cadherin is the loading control for membrane extracts. N=7 for total protein; N=8 for membrane protein; N=8 for intracellular protein. *p<0.05 vs egative control (NC) group. #p<0.05 vs TG group. NC, AAV9 virus carrying dCas9-SAM system with control sgRNA; Dys, AAV9 virus carrying dCas9-SAM system with sgRNA targeting dystrophin promoter. (**B**) The mRNA level of *Scn5a* by real-time PCR (N=10–12) and the total, membrane and intracellular protein levels of Nav1.5 by western blot. N-cadherin is the loading control for membrane extracts. N=10 for total protein; N=5 for membrane protein; N=6 for intracellular protein. *p<0.05 vs NC group. #p<0.05 vs TG group. (**C**) Representative traces, I-V curves and kinetics of peak $I_{Na}$ currents. N=7–20 cells; N=3 mice of WT + NC; N=3 mice of WT + Dys; N=3 mice of TG + NC; N=4 mice of TG + Dys.

*Figure 7 continued on next page*

*Figure 7 continued*

*p<0.05 vs NC group. #p<0.05 vs TG group. (**D**) Conduction velocity of perfused hearts by optical mapping recordings. N=6–9. *p<0.05 vs NC group. #p<0.05 vs TG group. p-values were determined by unpaired t-test. (**E**) Ventricular fibrillation (VF) induced by S1S1 pacing in perfused hearts. N=7–10. (**F**) VF induced by S1S2 pacing in intact mice. N=10–13. The data are analyzed by one-way ANOVA followed by Tukey's post-hoc analysis.

The online version of this article includes the following source data and figure supplement(s) for figure 7:

**Source data 1.** Source data for *Figure 7*.

**Source data 2.** Uncropped and labeled gels for *Figure 7A*.

**Source data 3.** Raw unedited gels for *Figure 7A*.

**Source data 4.** Uncropped and labeled gels for *Figure 7B*.

**Source data 5.** Raw unedited gels for *Figure 7B*.

**Figure supplement 1.** Schematic model for the construction of adeno-associated virus 9 (AAV9) virus carrying the dCas9-SAM system (**A**) to activate the transcription of dystrophin and tail-vein injection to mice (**B**).

---

Care and Use of Laboratory Animals published by the US National Institutes of Health (NIH Publication No. 85–23, revised 1996).

## Neonatal cardiomyocytes preparation

Neonatal cardiomyocytes were isolated from 3-day-old mice in accordance with the following procedures. Briefly, after dissection, hearts were washed and minced in 0.25% trypsin. Pooled cell suspensions were centrifuged and resuspended in Dulbecco's modified Eagle's medium (DMEM Hyclone, USA) supplemented with 10% fetal bovine serum, 100 U/ml penicillin and 100 µg/ml streptomycin. The suspension was incubated in culture flasks for 90 min, which makes fibroblasts preferentially adhere to the bottom of the culture flasks. Neonatal cardiomyocytes were removed from the culture flasks and the medium was changed. Cell cultures were incubated for 48 hr at 37 °C in a humidified atmosphere of 95% oxygen and 5% carbon dioxide before any experimentation.

## Generation of cardiac myocyte-specific *lncDach1* overexpressing mice

Cardiomyocyte-specific *lncDach1* overexpressing mice driven by murine *Myh6* promoter on a C57BL/6 background was generated by Biocytogen Co., Ltd (Beijing, China) as demonstrated in a previous study (*Cai et al., 2019*).

## Generation of cardiomyocyte-specific *lncDach1* knockout mice

*LncDach1* conditional KO mice (*lncDach1* Flox/Flox) were generated by using CRISPR/Cas9 technique on C57BL/6 background mice by Biocytogen Co., Ltd (Beijing, China) as demonstrated in a previous study (*Cai et al., 2019*).

## Construction of adeno-associated virus 9 (AAV9) carrying deactivated clustered regularly interspaced short palindromic repeats associated protein 9 nuclease- synergistic activation mediator(dCas9-SAM) system to activate the transcription of dystrophin

AAV9 carrying dCas9-SAM system to activate the transcription of dystrophin was constructed as reported previously with brief modifications (*Maeder et al., 2013*). The sgRNA targeting on the promoter region of dystrophin was designed and cloned into the multiple cloning site of plasmid GV639 (EFS-NLS-dSaCas9-NLS-VP64-bGHpA-U6). The constructed plasmid was packaged into the AAV9 virus. The sequence of sgRNA is: 5'- CGCTTCCGCGGCCCGTTCAA –3'; The mock-sgRNA target sequence (5'-CGCTTCCGCGGCCCGTTCAA –3') was used as a negative control. The obtained AAV9 virus volume was administered into C57BL/6 mice via tail vein injection at $1\times10^{11}$ genome containing particles (GC)/animal in 100 µl.

## Construction of adenovirus carrying cF- *lncDach1* and in vivo gene delivery

Adenovirus vectors carrying cF- *lncDach1* (OE- cF- *lncDach1*) and a negative control (NC) and a CAG promoter conjugated with green fluorescent protein (GFP) were constructed by Genechem Co., Ltd.

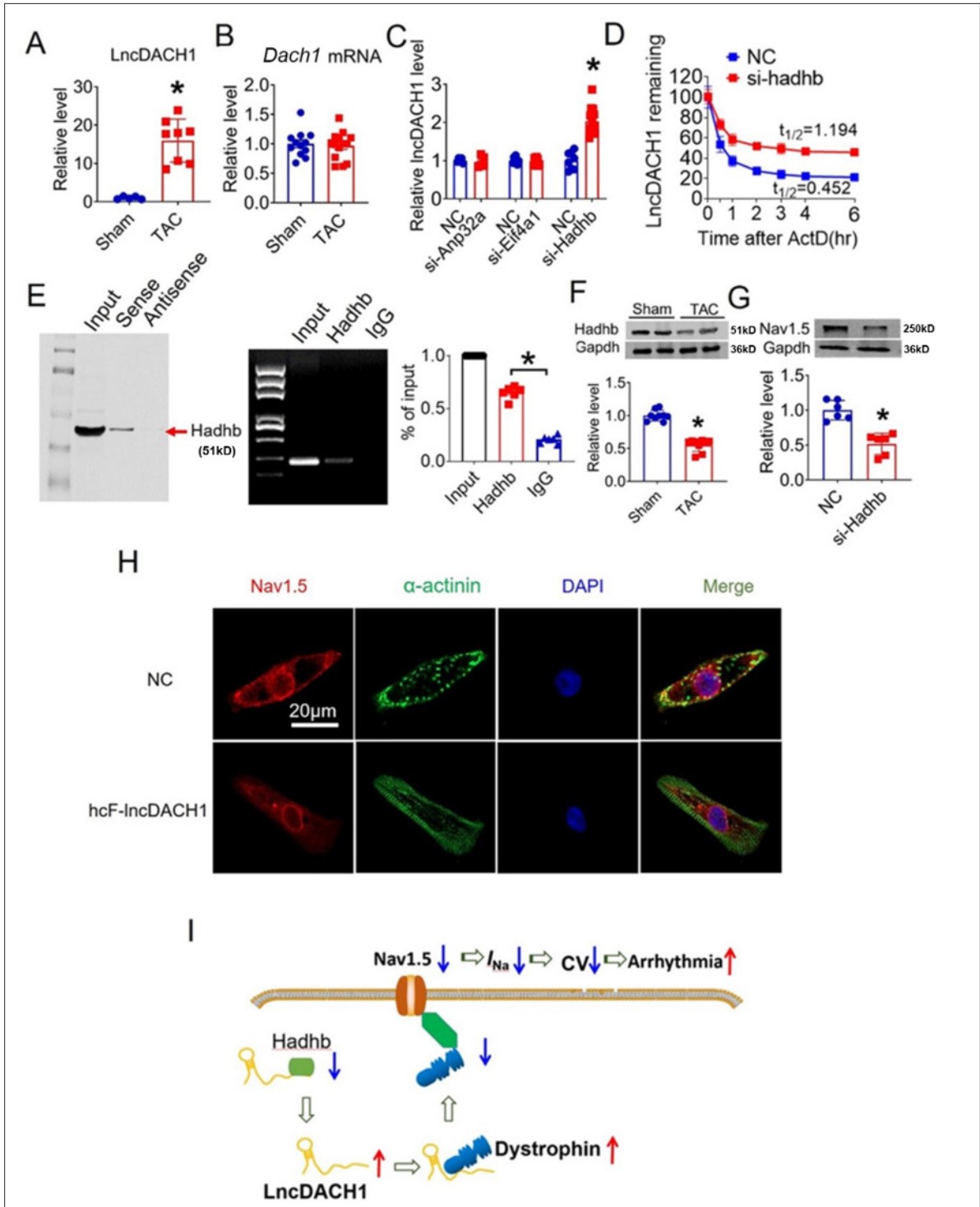

**Figure 8.** Hadhb binds to lncRNA-Dachshund homolog 1 (*lncDach1*) and promotes its decay. (**A**) The expression level of *lncDach1* in the hearts of transaortic constriction (TAC) mice. N=5–8. *p<0.05 by unpaired t-test. (**B**) The mRNA level of *Dach1* in the hearts of TAC mice. N=12–14. (**C**) Effects of siRNAs for *Anp32a*, *Eif4a1,* and *Hadhb* on the expression of *lncDach1*. N=5–12 from three independent cultures. * p<0.05 by unpaired t-test. (**D**) The effects of *Hadhb* siRNA on the decay of *lncDach1*. N=6–15 from three independent cultures. (**E**) Blotting of hadhb pulled-down by *lncDach1*, and precipitation of *lncDach1* by anti-hadhb antibody. N=4. *p<0.05 vs IgG by one-way ANOVA followed by Tukey's post-hoc analysis. (**F**) The effects of heart failure on the protein expression of hadhb. N=9. *p<0.05 vs sham group. p-values were determined by unpaired t-test. (**G**) The effects of *Hadhb* siRNA on the protein expression of Nav1.5, N=6. *p<0.05 vs negative control (NC) group. p-values were determined by unpaired t-test. (**H**) *LncDACH1*

*Figure 8 continued on next page*

*Figure 8 continued*

inhibits Nav1.5 in human induced pluripotent stem cells (iPS) differentiated cardiomyocytes. (**I**) Schematic summary of the signaling pathway of *lncDACH1* and arrhythmia.

The online version of this article includes the following source data and figure supplement(s) for figure 8:

**Source data 1.** Source data for *Figure 8A–G*.

**Source data 2.** Uncropped and labeled gels for *Figure 8E*.

**Source data 3.** Raw unedited gels for *Figure 8E*.

**Source data 4.** Uncropped and labeled gels for *Figure 8F*.

**Source data 5.** Raw unedited gels for *Figure 8F*.

**Source data 6.** Uncropped and labeled gels for *Figure 8G*.

**Source data 7.** Raw unedited gels for *Figure 8G*.

**Figure supplement 1.** Fluorescence intensity of Nav1.5 in human induced pluripotent stem cells (iPS) differentiated cardiomyocytes overexpressing cF-*lncDACH1*.

**Figure supplement 1—source data 1.** Source data for *Figure 8—figure supplement 1*.

**Figure supplement 2.** Effects of cF-*lncDACH1* overexpression on sodium current and Vmax of APD in human induced pluripotent stem cells (hiPS)-CMs.

**Figure supplement 2—source data 1.** Source data for *Figure 8—figure supplement 2B, D*.

**Figure supplement 3.** Ubiquitination of Nav1.5 in the Primary cardiomyocytes of lncRNA-Dachshund homolog 1 (*lncDach1*) transgenic overexpression and knockout.

**Figure supplement 3—source data 1.** Uncropped and labeled gels for *Figure 8—figure supplement 3A*.

**Figure supplement 3—source data 2.** Raw unedited gels for *Figure 8—figure supplement 3A*.

**Figure supplement 3—source data 3.** Uncropped and labeled gels for *Figure 8—figure supplement 3B*.

**Figure supplement 3—source data 4.** Raw unedited gels for *Figure 8—figure supplement 3B*.

(Shanghai, China). OE- cF- *lncDach1*, control constructs at $1\times10^9$ genome containing particles (GC)/animal in 100 μl volume was administered into C57BL/6 mice with body weights ranging from 18~22 g via tail vein injection. Seventy-two hours after injection, the mice were subjected to further study.

## Construction of adenovirus carrying *lncDach1*, *lncDach1* siRNA, conserved fragment of human *lncDach1* and infection

Adenovirus vectors carrying *lncDach1* (OE-*lncDach1*), a short RNA fragment for silencing *lncDach1* (Si- *lncDach1*) or conserved fragment of human *lncDach1* (hcF- *lncDach1*) and a CAG promoter conjugated with green fluorescent protein (GFP) were constructed by Genechem Co., Ltd. (Shanghai, China). Neonatal cardiomyocytes were infected with adenovirus for 48 hr, and then subjected to subsequent study.

## Transfection of *Hadhb*, *Eif4a1*, and *Anp32* siRNA

siRNA for *Hadhb* (si*Hadhb*), *Eif4a*1(si*Eif4a1*), *Anp32*(si*Anp32*), and a scrambled negative control RNA (siNC) were synthesized by Generalbiol (Chuzhou, Anhui, China). These siRNAs were transfected at a final concentration of 100 nM into NMVMs using the X-treme GENE Transfection Reagent (Roche, Indianapolis, USA) according to the manufacturer's protocols. The cardiomyocytes were collected for total RNA isolation or protein purification.

## Induction of ventricular arrhythmia

C57BL/6 mice were anesthetized with 2,2,2-tribromoethanol (200 mg/kg, i.p.). An octapolar electrophysiological catheter (1.1 F, SciSense Inc, Canada) was inserted into the right ventricle via the jugular vein. Intracardiac pacing was performed using an automated stimulator interfaced with the data acquisition system (GY6000; HeNan HuaNan Medical Science & Technology Ltd., Zhengzhou, China). The surface recording electrode was fixed on the LV epicardium to record pseudo-ECG. Inducibility of VT was determined by applying a train of ten consecutive electrical pulses with a coupling interval of 80 ms (S1), followed by two extra stimuli (S2 and S3) at coupling intervals of 2 ms, respectively.

Successful induction of VT was defined as the appearance of rapid nonsinus rhythm ventricular activations lasting for three beats or more.

## Optical mapping recording

Mice were heparinized and euthanized by 2,2,2-tribromoethanol (200 mg/kg, intraperitoneal injection; Sigma, St Louis, MO, USA). The heart was isolated and Langendorff perfused with Tyrode's solution (NaCl 128.2 mM, $CaCl_2 \cdot 2H_2O$ 1.3 mM, KCl 4.7 mM, $MgCl_2 \cdot 6H_2O$ 1.85 mM, $NaH_2PO_4 \cdot 2H_2O$ 1.19 mM, $Na_2CO3$ 20 mM, and glucose 11.1 mM; pH 7.35) at 37 °C. After 10 min of stabilization, the hearts were stained with RH237 (10 µM) for membrane voltage (Vm) mapping. The dye was excited at 710 nm using a monochromatic light-emitting device. The fluorescence was filtered and recorded simultaneously with a MiCAM05 CMOS camera (SciMedia, USA) at 1 ms/frame and 100x100 pixels with a spatial resolution of 0.35×0.35 $mm^2$ per pixel. Blebbistatin (10 µM, Selleckchem, Houston, TX, USA) was used to inhibit motion artifacts during optical mapping.

## Experimental protocol of optical mapping

A pair of hook bipolar electrodes was inserted into the apex of the heart for pacing. A pseudo-ECG was obtained with widely spaced bipolar electrodes to determine ventricular rhythm. The ventricles were initially paced at a constant pacing cycle length (PCL) of 200 ms. The PCLs were progressively shortened (200, 100, 60, 40, 30, 20 ms) with a duration of 1–2 s until VT was induced or the loss of 1:1 capture of the ventricles. Optical recording was performed after 20 beats of stable pacing at each PCL. Optical recordings were then performed during VT.

## Patch-clamp recording

Whole-cell configuration of the patch-clamp technique was used to record peak $I_{Na}$ and the inward rectifier K⁺ current ($I_{K1}$). $I_{Na}$ recordings were performed at room temperature (22~23 °C) by using a MultiClamp 700B (Alembic Instruments) amplifier. Pipettes (tip resistance 1–2 MΩ) were filled with a solution containing (in mM): NaCl 5, $CaCl_2$ 2, $MgCl_2$ 2, CsCl 130, HEPES 10, EGTA 15, and MgATP 4 (pH 7.2 with CsOH). Myocytes were bathed with a solution containing (in mM): NaCl 25, $CaCl_2$ 2, $MgCl_2$ 2.5, CsCl 108.5, HEPES 10, $CoCl_2$ 2.5, and glucose 10 (pH 7.4 with CsOH). $I_{K1}$ was measured as a $Ba^{2+}$-sensitive steady-state current and treated with 300 µmol/L $Ba^{2+}$ recording at 37 C using a patch clamp amplifier (MultiClamp 700B). The external solution for recording $I_{K1}$ contained (in mM): NaCl 136, KCl 5.4, $NAH_2PO_4$ 0.33, $MgCl_2 \cdot 6H_2O$ 1, $CaCl_2$ 1.8, HEPES 5 and glucose 10 (pH 7.37 with NaOH). The pipette solution for recording $I_{K1}$ contained (in mM): KCl 20, K-aspartate 110, HEPES 5, $MgCl_2$ 1, $Na_2ATP$ 5, and EGTA 10 (pH 7.20 with KOH).

## Differentiation of human induced pluripotent stem cells(hiPSCs) to cardiomyocytes

Undifferentiated hiPS cells (AC-iPSC) were purchased from NC5 (Help Stem Cell Innovations, NC2001) and cultured on Matrigel-coated plates in an E8 medium (CA1001500, CELLAPY). The iPSCs were generated from PBMCs. Flow cytometry analysis and immunofluorescence analysis show that the iPSCs demonstrated high expression levels of pluripotency markers (TRA-1–60, SSEA4, and OCT4). The iPSCs tested negative for mycoplasma contamination. Differentiation basal medium composed of RPMI1640 medium (C11875500BT, Thermo Fisher Scientific) and B27 minus insulin (A1895601, Thermo Fisher Scientific) was used to induce cardiomyocyte differentiation. Specifically, the 70~80% confluent hiPSCs were incubated in differentiation basal medium added with CHIR-99021 (HY-10182, MCE) for 1 d and Wnt-C59 (S7037, Selleck Chemicals) for 2 d. Then, the cells were cultured in RPMI1640 basal medium containing B27 (17504044, Thermo Fisher Scientific), which was replaced with fresh medium every 1~2 d. Beating cells were observed after 8 d of differentiation induction and used for further study.

## Construction of truncated *LncDach1* fragment plasmids

The sequence of *lncDach1* was divided into five fragments. The cDNA of each fragment was inserted into the pCDNA3.1, respectively. The first 417 nts of the entire sequences were cut off and constructed as fragment-a (418–2085 nts). Another 417 nts was cut off to generate fragment-b (835–2085 nts). The

third 417 nts was cut off to generate fragment-c (1251–2085 nts). Fragment-d is from 835 to 1668 nts, and fragment-e is from 835 to 1251 nts.

## Isolation of cardiac myocytes

Hearts were rapidly excised, cannulated, and perfused with $Ca^{2+}$-free Tyrode solution (in mM): NaCl 137, KCl 5.4, $NaH_2PO_4$ 0.16, glucose 10, $CaCl_2$ 1.8, $MgCl_2$ 0.5, HEPES 5.0, and $NaHCO_3$ 3.0 (pH 7.4 adjusted with NaOH) for 5 min. The heart was then perfused with a solution containing collagenases B and D (Roche) and protease XIV (Sigma) until digestion was complete. Tissue was dissociated using forceps, and dissociated left ventricular cardiomyocytes were gradually exposed to $Ca^{2+}$ (from 50 to 500 µM over 40 min) and plated in culture chambers for further studies.

## Immunocytochemistry of isolated mouse ventricular myocytes

Cardiomyocytes were fixed for 10 min with 4% paraformaldehyde in PBS, and then washed in PBS for 10 min (two times). The cells were permeabilized with 0.5% Tween 20 for 30 min. After being washed out with PBS for 10 min (three times), cardiomyocytes were incubated with primary antibodies against Nav1.5 (ASC005, Alomone, 1;200) and dystrophin (MANDYS8, SIGMA, 1;300) overnight at 4 °C. Following washout with PBS (10 min, three times), cells were incubated with secondary antibodies for 1 hr and washed with PBS (10 min, three times). The coverslips were mounted onto frosted slides in a solution composed of 90% FluorSave Reagent (Calbiochem, La Jolla, CA, USA) and 10% 10 X PBS.

## Fluorescent in situ hybridization (FISH)

In situ hybridization was performed with a FISH Kit (RiboBio, Guangzhou, China). Briefly, isolated cardiomyocytes were fixed in 4% formaldehyde at 4 °C for 10 min and dried out on the slides at room temperature (RT). The slides were rinsed and permeabilized with 0.5% Triton-100 in PBS at RT for 30 min, washed with PBS solution, and prehybridized at 37 °C for 30 min before hybridization. The prehybridized slides were then incubated with lncRNA-probe in a hybridization solution at 37 °C for 16 hr. After hybridization, the slides were washed six times with prewarmed wash buffer and PBS solution. Finally, the slides were counterstained with DAPI and visualized using a confocal laser-scanning microscope (Zeiss 800, Germany).

## Quantitative real-time RT-PCR

Total RNA was extracted by using Trizol reagent (Invitrogen, USA) according to the manufacturer's protocol. Total RNA (0.5 µg) was reverse transcribed by using the TransScript reverse transcriptase (GMO technology, Beijing) to obtain cDNA. The RNA levels were determined using SYBR Green I incorporation method on ABI 7500 fast Real-Time PCR system (Applied Biosystems, USA), The expression levels of mRNA were calculated using the comparative cycle threshold (Ct) method ($2-\Delta\Delta Ct$). Each data point was then normalized to *Actin* as an internal control in each sample. The final results are expressed as fold changes by normalizing the data to the values from control subjects. Primers used in this study were listed in *Supplementary file 1* .

## Western blot

The total, membrane, and intracellular protein samples were extracted from cardiac tissues of C57BL/6 mice for immunoblotting analysis. Total protein was collected with the treatment of RIPA lysis buffer (Beyotime, Beijing, China) and a protease inhibitor cocktail (Roche, Basel, Switzerland) at 4 °C followed by centrifugation. Extraction of surface and intracellular proteins was conducted using the Surface and Intracellular Protein Reagent Kit (Cat#P0033; Beyotime, Shanghai, China) according to the manufacturer's instructions. Protein samples were fractionated by SDS-PAGE and then transferred to the PVDF membrane. The membranes were blocked in Tris-buffered saline containing 5% milk and then incubated with primary antibodies at 4 °C overnight. The primary antibodies include anti-Nav1.5 (ASC005, Alomone, 1:200), anti-dystrophin (MANDYS8, SIGMA, 1:500). The anti-β-actin (1:20000 dilution, 66009–1-Ig, Proteintech), and anti-N-cadherin antibody (Cat#ab76011, 1:5000; Abcam, Cambridge, UK) were used as internal controls. Western blot bands were captured on the Odyssey Infrared Imaging System (LI-COR Biosciences, USA) and quantified with Odyssey v1.2 software by measuring the band intensity (area ×OD) in each group. The band intensity was normalized to the internal control. All antibodies were diluted in PBS buffer.

## RNA pulldown and immunoblotting

The RNA pull-down was performed as described in the previous study (*Cai et al., 2019*). Briefly, Biotin-labeled, full-length *lncDach1* RNA and antisense RNA were prepared with the Biotin RNA Labeling Mix (Roche) and T7 RNA polymerase (Roche). Biotinylated RNAs were treated with RNase-free DNase I (Invitrogen) and purified on G-50 Sephadex Quick Spin columns (Roche). Biotinylated RNA (17 µg) was heated to 65 °C for 10 min and slowly cooled to 4 °C. Then the RNA was mixed with tissue extracts in pulldown buffer supplemented with tRNA (0.1 µg/µl) and incubated at 4 °C for 2 hr. Washed Streptavidin agarose beads (60 µl, Invitrogen) were added to each binding reaction and further incubated at 4 °C for 1 hr. Beads were washed briefly five times in pulldown buffer and boiled in SDS buffer, and the retrieved protein was visualized by immunoblotting.

## RNA immunoprecipitation (RIP)

RIP experiments were performed by using a Magna RIPTM RNA-Binding Protein Immunoprecipitation Kit (Millipore, USA) as previously reported (*Cai et al., 2019*). Briefly, heart tissue was pieced and lysed in 220 µl of lysis buffer containing protease inhibitors and RNase Inhibitor and centrifuged at 14,000×g for 10 min. The supernatants were incubated with anti-dystrophin, anti-hadhb, and anti-rabbit IgG antibody for overnight at 4 °C with gentle rotation. Protein G magnetic beads (50 µl) were added and incubated at RT with gentle rotation for 3 hr. RNA was extracted with 400 µl phenol:chloroform:isoamyl alcohol (125:24:1, pH = 4.3) according to the manufacturer's instructions before quantitation by RT-qPCR.

## Mouse models of heart failure (HF) by transaortic constriction (TAC) and by coronary artery ligation

Mice were randomly divided into sham and TAC groups. In each group, mice were anesthetized with 2,2,2-tribromoethanol (200 mg/kg, i.p.) for the TAC model. The animal was orally intubated with a 20-gauge tube, and ventilated (mouse ventilator, UGO BASILE, Biological Research Apparatus, Italy) at the respiratory rate of 100 breaths/min with a tidal volume of 0.3 ml. The transverse aorta was constricted by a 6–0 silk suture ligature tied firmly against a 27-gauge needle between the carotid arteries. Then, the needle was promptly removed to yield a constriction of 0.4 mm in diameter. For sham group mice, the animals received the same procedures without aorta constriction.

## Statistical analysis

Data are expressed as mean ± SEM. Statistical analysis was performed using unpaired Student's t-test or One-Way Analysis of Variance (ANOVA) followed by Tukey's post-hoc analysis. A $p < 0.05$ was considered statistically different.

# Acknowledgements

G Xue, Y Zhang, J Yang, performed experiments, analyzed data, and prepared the manuscript. Y Yang, R Zhang, D Li, T Tian, J Li, X Zhang, C Li, X Li, J Yang, K Shen, Y Guo, X Liu, and G Yang helped perform experiments and collect data. B Yang and Y Lu oversaw the project and proofread the manuscript. Z Pan designed the project, oversaw the experiments and prepared the manuscript. This work was supported by National Key R&D Program of China (2017YFC1307404 to Z P), National Natural Science Foundation of China (82070344, 81870295 to Z P 81730012, 81861128022 to B Y), Heilongjiang Touyan Innovation Team Program, and CAMS Innovation Fund for Medical Sciences (CIFMS, 2019-I2M-5–078 to B Y).

## Additional information

### Funding

| Funder | Grant reference number | Author |
|---|---|---|
| National Key Research and Development Program of China | | Genlong Xue<br>Yang Zhang<br>Ying Yang<br>Ruixin Zhang<br>Desheng Li<br>Tao Tian<br>Jialiang Li<br>Xiaofang Zhang<br>Changzhu Li<br>Xingda Li<br>Jiqin Yang<br>Kewei Shen<br>Yang Guo<br>Xuening Liu<br>Guohui Yang<br>Lina Xuan<br>Hongli Shan<br>Yanjie Lu<br>Yang Baofeng<br>Zhenwei Pan<br>Jiming Yang |
| National Natural Science Foundation of China | | Genlong Xue<br>Yang Zhang<br>Ying Yang<br>Ruixin Zhang<br>Desheng Li<br>Tao Tian<br>Jialiang Li<br>Xiaofang Zhang<br>Changzhu Li<br>Xingda Li<br>Jiqin Yang<br>Kewei Shen<br>Yang Guo<br>Xuening Liu<br>Guohui Yang<br>Lina Xuan<br>Hongli Shan<br>Yanjie Lu<br>Yang Baofeng<br>Zhenwei Pan<br>Jiming Yang |
| CIFMS | | Genlong Xue<br>Yang Zhang<br>Ying Yang<br>Ruixin Zhang<br>Desheng Li<br>Tao Tian<br>Jialiang Li<br>Xiaofang Zhang<br>Changzhu Li<br>Xingda Li<br>Jiqin Yang<br>Kewei Shen<br>Yang Guo<br>Xuening Liu<br>Guohui Yang<br>Lina Xuan<br>Hongli Shan<br>Yanjie Lu<br>Yang Baofeng<br>Zhenwei Pan<br>Jiming Yang |

| Funder | Grant reference number | Author |
| --- | --- | --- |
| National Key R&D Program of China | 2017YFC1307404 | Zhenwei Pan |
| National Natural Science Foundation of China | 82070344 | Zhenwei Pan |
| National Natural Science Foundation of China | 81870295 | Zhenwei Pan |
| National Natural Science Foundation of China | 81730012 | Zhenwei Pan |
| National Natural Science Foundation of China | 81861128022 | Zhenwei Pan |
| Heilongjiang Touyan Innovation Team Program, and CAMS Innovation Fund for Medical Sciences | 2019-I2M-5–078 | Yang Baofeng |
| Dalian Medical Key Specialty Climbing Program Scientific Research Project | 2022DF014 | Genlong Xue |
| Liaoning Provincial Natural Science Foundation Project | 2023-BS-166 | Genlong Xue |

The funders had no role in study design, data collection and interpretation, or the decision to submit the work for publication.

### Author contributions

Genlong Xue, Conceptualization, Data curation, Methodology, Writing – original draft, Project administration; Jiming Yang, Data curation, Validation, Project administration, Writing – review and editing; Yang Zhang, Supervision, Project administration, Writing – review and editing; Ying Yang, Tao Tian, Supervision, Validation, Investigation; Ruixin Zhang, Yanjie Lu, Supervision, Validation, Writing – review and editing; Desheng Li, Jialiang Li, Xiaofang Zhang, Jiqin Yang, Kewei Shen, Yang Guo, Xuening Liu, Guohui Yang, Lina Xuan, Hongli Shan, Supervision, Validation; Changzhu Li, Xingda Li, Formal analysis, Supervision, Validation; Yang Baofeng, Resources, Supervision, Funding acquisition, Validation, Project administration, Writing – review and editing; Zhenwei Pan, Conceptualization, Resources, Data curation, Funding acquisition, Project administration, Writing – review and editing

### Author ORCIDs

Jiming Yang http://orcid.org/0000-0003-0007-5152
Desheng Li https://orcid.org/0000-0002-4884-745X
Zhenwei Pan https://orcid.org/0000-0002-1011-0954

### Ethics

The study was conducted in accordance with the Declaration of Helsinki, all experimental procedures complied with IRB of College of Pharmacy,Harbin Medical University (Approval Number: IRB2004821).

Reviewer #1 (Public Review): https://doi.org/10.7554/eLife.89690.4.sa1
Reviewer #2 (Public Review): https://doi.org/10.7554/eLife.89690.4.sa2
Reviewer #3 (Public Review): https://doi.org/10.7554/eLife.89690.4.sa3
Author response https://doi.org/10.7554/eLife.89690.4.sa4

## Additional files

### Supplementary files

MDAR checklist

Supplementary file 1. List of primers used in this study.

Supplementary file 2. Alignment of murine lncDACH1.

### Data availability

All data generated or analysed during this study are included in the manuscript source data and supplementary files.

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
