## [Editor Report · eLife assessment]

This study presents an **important** contribution to cardiac arrhythmia research by demonstrating long noncoding RNA Dachshund homolog 1 (lncDACH1) tunes sodium channel functional expression and affects cardiac action potential conduction and rhythms. The evidence supporting the major claims are **convincing**. The work will be of broad interest to cell biologists and cardiac electrophysiologists.

---

## [Referee Report · Reviewer #1 (Public Review)]

Summary:

In this study, the authors show that a long-non coding RNA lncDACH1 inhibits sodium currents in cardiomyocytes by binding to and altering the localization of dystrophin. The authors use a number of methodologies to demonstrate that lncDACH1 binds to dystrophin and disrupts its localization to the membrane, which in turn downregulates NaV1.5 currents. Knockdown of lncDACH1 upregulates NaV1.5 currents. Furthermore, in heart failure, lncDACH1 is shown to be upregulated which suggests that this mechanism may have pathophysiological relevance.

Strengths:

(1) This study presents a novel mechanism of Na channel regulation which may be pathophysiologically important.

(2) The experiments are comprehensive and systematically evaluate the physiological importance of lncDACH1.

---

## [Referee Report · Reviewer #2 (Public Review)]

This manuscript by Xue et al. describes the effects of a long noncoding RNA, lncDACH1, on the localization of Nav channel expression, the magnitude of INa, and arrhythmia susceptibility in the mouse heart. Because lncDACH1 was previously reported to bind and disrupt membrane expression of dystrophin, which in turn is required for proper Nav1.5 localization, much of the findings are inferred through the lens of dystrophin alterations.

The results report that cardiomyocyte-specific transgenic overexpression of lncDACH1 reduces INa in isolated cardiomyocytes; measurements in the whole heart show a corresponding reduction in conduction velocity and enhanced susceptibility to arrhythmia. The effect on INa was confirmed in isolated WT mouse cardiomyocytes infected with a lncDACH1 adenoviral construct. Importantly, reducing lncDACH1 expression via either a cardiomyocyte-specific knockout or using shRNA had the opposite effect: INa was increased in isolated cells, as was conduction velocity in the heart. Experiments were also conducted with a fragment of lnDACH1 identified by its conservation with other mammalian species. Overexpression of this fragment resulted in reduced INa and greater proarrhythmic behavior. Alteration of expression was confirmed by qPCR.

The mechanism by which lnDACH1 exerts its effects on INa was explored by measuring protein levels from cell fractions and immunofluorescence localization in cells. In general, overexpression was reported to reduce Nav1.5 and dystrophin levels and knockout or knockdown increased them.

The strengths of this manuscript include convincing evidence of a link between lncDACH1 and Na channel function. The identification of a lncDACH1 segment conserved among mammalian species is compelling. The observation that lncDACH1 is increased in a heart failure model and provides a plausible hypothesis for disease mechanism.

---

## [Referee Report · Reviewer #3 (Public Review)]

Summary:

In this manuscript, the authors report the first evidence of Nav1.5 regulation by a long noncoding RNA, LncRNA-DACH1, and suggest its implication in the reduction in sodium current observed in heart failure. Since no direct interaction is observed between Nav1.5 and the LncRNA, they propose that the regulation is via dystrophin and targeting of Nav1.5 to the plasma membrane.

Strengths:

(1) First evidence of Nav1.5 regulation by a long noncoding RNA.

(2) Implication of LncRNA-DACH1 in heart failure and mechanisms of arrhythmias.

(3) Demonstration of LncRNA-DACH1 binding to dystrophin.

(4) Potential rescuing of dystrophin and Nav1.5 strategy.

---

## [Author Response]

The following is the authors’ response to the previous reviews.

**eLife assessment**
This study presents an important contribution to cardiac arrhythmia research by demonstrating long noncoding RNA Dachshund homolog 1 (lncDACH1) tunes sodium channel functional expression and affects cardiac action potential conduction and rhythms. The evidence supporting the major claims are solid. The work will be of broad interest to cell biologists and cardiac electrophysiologists.
**Public Reviews:**

**Reviewer #1 (Public Review):**
Summary:In this study, the authors show that a long-non coding RNA lncDACH1 inhibits sodium currents in cardiomyocytes by binding to and altering the localization of dystrophin. The authors use a number of methodologies to demonstrate that lncDACH1 binds to dystrophin and disrupt its localization to the membrane, which in turn downregulates NaV1.5 currents. Knockdown of lncDACH1 upregulates NaV1.5 currents. Furthermore, in heart failure, lncDACH1 is shown to be upregulated which suggests that this mechanism may have pathophysiological relevance.Strengths:(1) This study presents a novel mechanism of Na channel regulation which may be pathophysiologically important.(2) The experiments are comprehensive and systematically evaluate the physiological importance of lncDACH1.
**Reviewer #2 (Public Review):**
This manuscript by Xue et al. describes the effects of a long noncoding RNA, lncDACH1, on the localization of Nav channel expression, the magnitude of INa, and arrhythmia susceptibility in the mouse heart. Because lncDACH1 was previously reported to bind and disrupt membrane expression of dystrophin, which in turn is required for proper Nav1.5 localization, much of the findings are inferred through the lens of dystrophin alterations.The results report that cardiomyocyte-specific transgenic overexpression of lncDACH1 reduces INa in isolated cardiomyocytes; measurements in whole heart show a corresponding reduction in conduction velocity and enhanced susceptibility to arrhythmia. The effect on INa was confirmed in isolated WT mouse cardiomyocytes infected with a lncDACH1 adenoviral construct. Importantly, reducing lncDACH1 expression via either a cardiomyocyte-specific knockout or using shRNA had the opposite effect: INa was increased in isolated cells, as was conduction velocity in heart. Experiments were also conducted with a fragment of lnDACH1 identified by its conservation with other mammalian species. Overexpression of this fragment resulted in reduced INa and greater proarrhythmic behavior. Alteration of expression was confirmed by qPCR.The mechanism by which lnDACH1 exerts its effects on INa was explored by measuring protein levels from cell fractions and immunofluorescence localization in cells. In general, overexpression was reported to reduce Nav1.5 and dystrophin levels and knockout or knockdown increased them.The strengths of this manuscript include convincing evidence of a link between lncDACH1 and Na channel function. The identification of a lncDACH1 segment conserved among mammalian species is compelling. The observation that lncDACH1 is increased in a heart failure model and provides a plausible hypothesis for disease mechanism.One limitation of the fractionation approach is the uncertain disposition of Na channel protein deemed "cytoplasmic." It seems likely that the membrane fraction includes ER membrane. The signal may reasonably be attributed to Na channel protein in stalled transport vesicles, or alternatively in stress granules, but this was not directly addressed.
**Reviewer #3 (Public Review):**
Summary:In this manuscript, the authors report the first evidence of Nav1.5 regulation by a long noncoding RNA, LncRNA-DACH1, and suggest its implication in the reduction in sodium current observed in heart failure. Since no direct interaction is observed between Nav1.5 and the LncRNA, they propose that the regulation is via dystrophin and targeting of Nav1.5 to the plasma membrane.Strengths:(1) First evidence of Nav1.5 regulation by a long noncoding RNA.(2) Implication of LncRNA-DACH1 in heart failure and mechanisms of arrhythmias.(3) Demonstration of LncRNA-DACH1 binding to dystrophin.(4) Potential rescuing of dystrophin and Nav1.5 strategy.Weaknesses:(1) The fact that the total Nav1.5 protein is reduced by 50% which is similar to the reduction in the membrane reduction questions the main conclusion of the authors implicating dystrophin in the reduced Nav1.5 targeting. The reduction in membrane Nav1.5 could simply be due to the reduction in total protein.
**Recommendations for the authors:**

**Reviewer #1 (Recommendations For The Authors):**
Weaknesses:(1) What is indicated by the cytoplasmic level of NaV1.5, a transmembrane protein?This is still confusing. Since Nav1.5 is an integral membrane protein, I am not sure what is really meant here by cytosolic fraction. From the workflow, it seems a separate organelle fraction is also collected. Is the amount of Nav1.5 in this fraction (which I assume includes for e.g. lysosome) also increased with lncDACH1? I recommend the authors to refer to the Nav channels not at the plasma membrane as 'intracellular' rather than 'cytoplasmic'.

Thanks for the insightful comment. We completely agree. Accordingly, we have changed “cytoplasmic” to “ intracellular“.

Line 226. "In consistent with the results" Perhaps unnecessary to have "in"

Thank you for the insightful comment. We have corrected it.

Line 228. Is it optimal or optical?

Sorry for the mistake, it should be optical. We have corrected it.

**Reviewer #3 (Recommendations For The Authors):**
I still have an issue with the total reduction in Nav1.5 which is about the same as the reduction in membrane and currents. The authors argue that there is an increase in cytoplasmic Nav1.5. However the controls that they provide for membrane and cytoplasmic fractions are not convincing.

Thank you for the insightful comment. We can not rule out the possibility that the reduction in membrane Nav1.5 maybe be due to the reduction in total protein. Our data indicates that the membrane and total protein levels of Nav1.5 were reduced by 50%. However, the intracellular Nav1.5 was not decreased, but increased in the hearts of lncDACH1-TG mice than WT controls, which indicates that the intracellular Nav1.5 failed to traffic to the membrane.